# Functional Anatomy of the Thoracic Limb of the Komodo Dragon (*Varanus komodoensis*)

**DOI:** 10.3390/ani13182895

**Published:** 2023-09-12

**Authors:** Michał Kępa, Anna Tomańska, Joanna Staszewska, Małgorzata Tarnowska, Joanna Klećkowska-Nawrot, Karolina Goździewska-Harłajczuk, Amadeusz Kuźniarski, Tomasz Gębarowski, Maciej Janeczek

**Affiliations:** 1Department of Biostructure and Animal Physiology, Faculty of Veterinary Medicine, Wrocław University of Environmental and Life Sciences, Kożuchowska St. 1, 51-631 Wrocław, Poland; 123526@student.upwr.edu.pl (M.K.); 123035@student.upwr.edu.pl (J.S.); joanna.kleckowska-nawrot@upwr.edu.pl (J.K.-N.); karolina.gozdziewska-harlajczuk@upwr.edu.pl (K.G.-H.); maciej.janeczek@upwr.edu.pl (M.J.); 2Division of Histology and Embryology, Department of Biostructure and Animal Physiology, Faculty of Veterinary Medicine, Wrocław University of Environmental and Life Sciences, Norwida St. 25, 50-375 Wrocław, Poland; malgorzata.tarnowska@upwr.edu.pl; 3Department of Prosthetic Dentistry, Faculty of Dentistry, Wrocław Medical University, Krakowska St. 26, 50-425 Wrocław, Poland; amadeusz.kuzniarski@umw.edu.pl

**Keywords:** anatomy, monitor lizards, Komodo dragon, thoracic limb

## Abstract

**Simple Summary:**

The Komodo dragon’s conservation and acquisition of detailed biological knowledge are considered contemporary priorities. The geographical location and behavior of this animal make research in the wild difficult. *Varanus komodoensis* is gigantic, endemic, and relict—a link between ancestors and modern monitor lizards. The anatomical adaptations of the biological functions made by the movement system in the thoracic limb have not been well documented yet. During dissection, the predominance of the saddle joints over the ball joints and the extensive antigravity functions of the biceps and triceps muscle groups were confirmed. Furthermore, a greater differentiation of the forearm muscles in relation to the pelvic limb musculature was also observed. These analogies may indicate adaptation for enhanced manual dexterity and orientation skills in the animal’s natural habitat, characterized by a robust and well-preserved physique. The muscles ensuring the rotation of the ulna relative to the humerus are also well developed, giving the animal a possibility to rest in physiological support, transferring the center of gravity towards the head, thus indicating functions and determining locomotive abilities. The anatomical description has many cognitive values—in both veterinary orthopedics and maintenance in zoos, biodiversity conservation, and reconstructing archaeological data about extinct animals.

**Abstract:**

Since the Komodo dragon has been included on The International Union for Conservation of Nature (IUCN) Red List of Threatened Species, it is crucial to know in detail its biology as there is a limited availability of research material on these animals—mainly those who died in zoos or whose remains were found in the wild. Anatomy is essential for understanding physiology, identification of diseases, adaptations in the environment, and behavior. In this dissection study, the relationship of individual anatomical structures was analyzed, the anatomy of the active and passive movement system of the thoracic limb was described, photographs were taken, and a radiographic examination was conducted. This species has its own differences, even within closely related lizard species. *Varanus komodoensis* possesses triceps muscles with three heads, and the wrist is extended with additional bones for greater flexibility of the hand. The muscles of the forelimb are analogous to the hind limb; however, they differ in the mass of individual muscles, especially those predisposed to perform the most important antigravity and locomotive functions.

## 1. Introduction

The Komodo dragon (*Varanus komodoensis* [Ouwens, 1912]; Komodo Monitor, Island Monitor [1]; id. Biawak komodo, Ora, Buayadarat, Biwak Rakasa [2,3]) is an endemic representative of the Indonesian archipelago. It inhabits various islands in Eastern Indonesia (including Komodo [4], Rinca [5], Flores [6], Nusa Kode and Gili Motang, Gili Dasami, and Padar [7,8]). Within its range, eight subpopulations [9] have been distinguished. It belongs to the Toxicofera clade of squamate reptiles of the family Varanidae (monitor lizards) [10,11]. Anguimorpha has significant locomotor diversity; there are species with reduced limbs and legless species, as well as quadrupedal and bipedal species. Among them, certain lineages have evolved specialized limbs that suit their characteristic niches. In the Varanidae group itself, limb specialization is closely linked to environmental conditions, size and sex, and the specific functions the limbs perform [12,13,14,15]. The Komodo dragon stands out as the largest representative of lizards in the world [16]. However, there are size variations between them [17] (mostly dependent on the habitat [18]), and it has also been proven that the body size of the Komodo dragon on Flores has not changed in the last 900,000 years [19]. It is a species that represents gigantism in the animal world, which is confirmed by morphological studies [19].

Unfortunately, the literature on the Komodo dragon’s detailed anatomy is rather insufficient. In general, the research on this species can be categorized into two temporal phases: from 1912 to 1970 and later. The first focuses on expeditions, a taxonomy of the species, measurements and obtaining data from the local population and individuals in zoos, with the consequent difficulties in breeding. In the second phase, attempts were made to determine the occurrence of the species in individual areas of the Indonesian islands and research expeditions were organized to evaluate the behavior. After 1971, studies yielded insights into growth patterns, blood parameters, sexual dimorphism, social behavior, and chromosomal evolution [2,8,20,21,22]. Nevertheless, the literature remains constrained, often with local purview. Species expert Surahya S. (Gadjah Mada University, Indonesia) suggested a lack of research continuity and succession (https://nasional.kompas.com/read/2011/02/04/13481641/~Sains~Biologi?page=all, accessed on 14 January 2023 [23]). Advanced muscle measurements were conducted by Hutchinson J. et al. at the Royal Veterinary College (UK) on a deceased male Komodo dragon from the London Zoo. However, data pertaining to the muscles of the thoracic limb has not yet been published (https://www.theatlantic.com/science/archive/2015/05/the-dragon-autopsy/393890/, accessed on 16 January 2023 [24]). Studies carried out at the Department of Anatomy, Physiology and Pharmacology, Bogor Agricultural Institute (IPB University, Bogor, Indonesia), were valuable in the analysis but also focused largely on the skeleton [25]. This insufficiency of intricate musculoskeletal insight served as the impetus for our investigative pursuit, with the objective of providing a comprehensive characterization of the forelimb’s muscular and skeletal structures. Moreover, another motivation behind this choice was the unique research materials accessible with the opportunity for skeletal system comparisons. This initiative originated from the species’ critical endangerment, along with the conjecture concerning plausible locomotive functional adaptations and their potential behavioral associations. This pursuit is underscored by the distinctiveness of this species, which also shares traits with its ancestors.

In this study, an anatomical examination of the forelimb was performed with dissection, macroscopic and microscopic observation, X-ray imaging and biometrics. Correlations with literature data on *V. komodoensis* and other animals were researched. The scope of this article includes a contextual framework (derived from an extensive literature review), accentuates the most important forelimb differentiation in the Anguimorpha group and provides detailed data on this issue among the Varanidae family, notably *V. komodoensis* and its close and extinct relatives. Anatomical, biometric and functional findings are appealing to a diverse readership. The structure of lizard limbs, however, has not only been of interest to biologists but also to engineers who have created a biorobotic machine based on a project inspired by the spreading limb of the Komodo dragon. This model mimics the nonfully horizontal orientation of the limb bone, transitioning it from a bent to a horizontal position [26]. This adaptation can refer to the habitat terrain and the lizard’s distinctive locomotion pattern, information-gathering mechanisms, survival strategies, food acquisition, shelter and intraspecies social interactions. Therefore, apart from the main purpose of anatomical description, this study underscores potential functional adaptations and their relevance to comparative biology.

### Active and Passive Movement System—Between Past and Present Research

Current research provides insight into the Komodo dragon, yielding utility not only in species preservation management [27] but also in evolutionary modeling [28,29]. Hocknull S. et al. undertook measurements on both fossilized and recently deceased Komodo dragon humeri (maximum width of the shaft and distal condyle). Regarding Australian fossil specimens originating from the Pliocene and Pleistocene Epochs, the remains did not represent significant differences compared to present-day individuals [19].

Modern and paleobiological data facilitate a heightened understanding of the biological significance of morphological limb variations [29,30]. The extensive research on squamates has unveiled the phylogenetic relationships in the development and structure of the limbs, which makes this group significantly diverse [28,31]. In some taxa, a complete limb reduction has occurred [12], albeit the morphological findings and molecular data do not consistently align [13]. Such a variety of limbed and limbless forms is observed across Anguimorpha. Comparative studies of modern squamates (lizards, snakes, and amphisbaenians) have mainly been based on morphological differences in the head and limbs. For instance, in the case of *Varanus*, the elongation of the neck is notably characterized by the doubling in size of vertebrae rather than their multiplication [14]. General commonalities were found in the neck myology of varanid lizards, snakes (*Trimeresurus*) and gekkotans. This region was variable depending on, among other factors, whether the organism had limbs or not. It demonstrates numerous similarities but also special features, for example, those that may directly result from pedomorphism [32]. Animals possessing limbs can be assigned to the normal-limbed group among squamates [33]. Some species of varanids, *Varanus gouldii*, *V. panoptes* and *V. giganteus*, have developed facultative bipedalism, augmenting maneuverability and acceleration [34,35]. These adaptations have been dated back to 110 million years ago [36]. While many lizards that have reduced their forelimbs have specialized hind limbs, the reverse scenario is exceedingly uncommon [37]. Among the squamates, a significant divergence in limb structure correlates with an occupied niche. This is evident in the reconfiguration of finger and phalange numbers, which manifests in gait types and manual activities [38]. Several lizards employ their forelimbs for social purposes, such as holding the female during copulation [33,39]; the very success of individual taxonomic groups hinges on their locomotor specializations in the environment [40], leading to navigational improvements [40,41].

Within the extensive array of adaptations, shared characteristics have been identified in monitor lizards [42]. For instance, the trapezius inserts on the suprascapular, and the episternocleidomastoid attaches to both the occipital bone and to the T-shaped clavicle [43]. The larger the species is, the longer the limb is in relation to the snout length; it is also thicker [43]. Compared to others, lizards have the lowest peak and average values of muscle moment arms (which is the joint action of individual muscles) for pulling the elbow to the back and, at the same time, the highest for external rotation of the humerus. This shed light on the interconnectedness between the locomotive functions of the skeleton and muscles. It also assigns specific roles for muscles (where an analogous muscle in another taxon may serve as a lifter rather than an abductor), and this may influence the placement of muscle attachments on bones, thereby affecting their structural composition. In cases of increased limb arching, a correlation is established with more proximal muscle attachment to the humerus, humeral slenderness, reduced epiphyses, and enhanced humeral curvature, which collectively brings the muscle attachment sites closer to the axis of rotation [44]. This interplay, coupled with alternations in joint mobility and osteological range of motion, contributes to significant differentiation of muscle fibers [45].

Current understanding of the functional anatomy of individual taxa remains limited, hindering the comprehensive interpretation of fossilized animals, in particular, to reconstruct soft tissues like muscles [17]. In order to systematize knowledge, such integration is necessary, which also includes physiology and genetic data [46].

With increasing body weight and size, there is inconsistent selective pressure for support and propulsion, which increases musculoskeletal stress, and it does not give us a real ability to scale muscle architecture based on small lizard species. This is a particular defiance of discovering the world’s largest lizard. Some locomotive muscle adaptations are species-specific. In the case of the pelvic limb of *V. komodoensis*, an increase in the muscle mass of the cross-sectional area of the key leg muscles was observed, which is correlated with the spreading posture, providing the opportunity to depart from the upright posture of giants [17,47]. Greater muscle mass in individual areas is associated with the need for organized manual skills of the animal, the versatility of movement (distal location), and the necessary traction (proximal location) [48].

In lizards and other animals with comparable locomotion, notable interrelations exist between the forelimb and hindlimb, corresponding to their habitats. The *flexor carpi radialis* muscle is distinctly different in arboreal and ground-dwelling lizards, allowing them to bend the wrist more strongly when climbing and grabbing plants [49]. The Komodo dragon is distinguished from other monitor lizards by the presence of a fused scapula to a coracoid bone, which is associated with a typically terrestrial habitat; the presence of a *crista deltoidea* enables this animal to dig actively, and two *crista pectoralis* enable juveniles to climb trees. The arrangement of the ulna and two pea-shaped wrist bones give it greater flexibility of movement compared to the Malayan water dragon (*Varanus salvator* Laurenti, 1768) [26,50]. Moreover, in the Varanidae family, the time and degree of epiphyseal atrophy varied, particularly pronounced in large species that are able to maintain active growth during ontogenesis [51]. These features are adaptations of the musculoskeletal system closely related to the animal’s posture and the way it moves.

We can also observe a similar gait in American alligators (*Alligator mississippiensis* [Daudin, 1801]). It was established that the pelvic limb has a greater load-bearing capacity, and the extensor muscles are stronger than in the forelimb. This is a significant difference confirming the theory that the strength-to-weight ratio decreases with increasing animal size. However, in the case of the key muscles responsible for the survival strategy of the species, they are stronger and more developed [52,53]. In the entire Varanidae family, this is visibly indicated in the muscles responsible for flexing the elbow, wrist flexors and humeral protractors. Furthermore, muscles exhibiting these parameters are more prevalent in the thoracic limb compared to the pelvic limb. This observation suggests a cranial shift in the center of gravity with varanid growth. In larger individuals, this is compensated by the development of pectoral muscles and a reduction in stride length [54,55]. Locomotor performance abilities are dependent on interrelationships between leg length and stride, muscle fiber type composition, aerobic capacity, cardiac output, etc. [56].

Komodo dragons exhibit intricate social and feeding behaviors, including ambushing, stalking, and ritualized courtship behaviors. They are capable of running at almost 30 km/h, climbing, and diving. An adult individual can cover distances of up to 10 km per day (with an average distance of 1.8 km). However, some data indicate even 18 km (detik.com). A defensive or offensive attack is shown by a stiff gait, a specific movement of the neck, and shifting body weight from left to right [57,58,59]. The way the lizards move has been observed in nature; every few meters, the lizard stops, while the back part of the body rests on the ground, and the front part of the body and the neck is straightened at a 45-degree angle. When eating, the Komodo dragon uses its massive forelimbs to propel its entire body, allowing it to pull away a large bite. This holding and pulling technique is unique [59]. Muscle functionality has been explored in *V. salvator*; limb support is largely generated from back muscle strength. They also perform postural functions when walking and running [60]. However, the difference between *V. komodoensis* and the aforementioned animals lies in their ecomorphological traits. Adult lizards of the Komodo dragon are strictly terrestrial [61].

## 2. Materials and Methods

### 2.1. Animals

The Department of Biostructure and Animal Physiology of Wrocław University of Environmental and Life Sciences (Poland) conducted an examination on a single specimen of a 7-year-old female Komodo dragon, weighing 30 kg and measuring 93 cm SVL (snout to vent length) [62]. It originated from the Wrocław Zoological Garden (Wrocław, Poland).

The skeletal system was examined using roentgenography (X-ray). Images were interpreted using the CR 7 VET scanner (iM3, Lane Cove, NSW, Australia) and Vet-Exam^plus^ 9.3.0 software (iM3 Pty Ltd., Lane Cove, NSW, Australia). The female’s weight was recorded as 30.0 kg, with an SLV (snout to vent length) of 93.6 cm, HL (head length) of 18.0 cm, and EOS (ear opening-snout distance) of 15.4 cm. The total length of the forelimb measured 28.7 cm, while HDL (hand length) was 6.9 cm, FAL (forearm length) was 9.8 cm, and AL (arm length) was 12.0 cm. Descriptions of the forelimb anatomical structures were documented, encompassing both skeletal components and muscles.

### 2.2. Anatomical and Physiological Evaluation

A better understanding of the position of the muscles on the limb before the dissection itself was achieved through comparison with the skeleton of another *V. komodoensis* specimen from the academic repository of the Archaeozoology Laboratory and the Nature Museum (Wrocław University of Environmental and Life Sciences, Wrocław, Poland). Photographic documentation of forelimbs was captured using an OLYMPUS camera with a 3.5 aperture (Panasonic G Vario 14–45 mm f/3.5–5.6 ASPH. M.O.I.S. lens, Shinjuku-Ku, Japan). The muscles of the thoracic limb and ligaments were dissected and described, provided with illustrations (Adobe Inc., 2019. Adobe Illustrator, San Jose, CA, USA). Precise measurements of key thoracic limb muscles were made. Histological preparations were subjected to the hematoxylin and eosin staining procedure (HE). Tissues were fixed, embedded in paraffin, and then sectioned into 7–10 µm sections using a microtome. The average fiber width of selected muscles of the main movement action was measured using Nikon Eclipse 80i, NIS-Elements software (© 2023 Nikon Instruments Inc., Melville, NY, USA). Comparative studies were carried out, and the muscular system of the thoracic limb was described in detail. The Latin nomenclature was adopted in accordance with Nomina Anatomica Veterinaria (NAV 2017) [63].

The research was conducted and based on pre-existing literature about reptiles. The species *Varanus priscus* (*Megalania prisca*, extinct) was considered just like the Australian species, as the latest reports indicate that the evolutionary line of the Komodo dragon emerged from them [64]. Additionally, the tree goanna (lace monitor, *Varanus varius*) was included due to its closest contemporary relationship based on mitochondrial DNA testing [65]. *Varanus salvator* was also considered; some differences have already been noticed, which result directly from the ecological niche and food acquisition strategies. Anatomical data on crocodiles (Crocodylidae) and their relatives (Alligatoridae) [21,53,66,67,68], as well as *Iguana i. iguana* [69], have been included, along with comparative data on other tetrapods [30,67,70]. The method of video observation and analysis was employed for the assessment of individual muscles and their functions.

## 3. Results

*V. komodoensis*’s scapula is conjoined with the coracoid bone. The scapular cartilage is thin, flexible, and vulnerable to damage. This structure is susceptible to damage during dissection and preparation of the skeleton, comparable to the vulnerability of the naviculars in the last phalanges. The Komodo dragon’s thoracic limb is joined by the extrinsic muscles of the forelimb (synsarcosis). The scapula not only articulates with the humerus but is also linked to the trunk, cranially closing the chest with the ventral process of the last cervical vertebra and the clavicle. The forelimb skeleton is supported by strong articulations; the diaphysis of the long bones is notably wider than the epiphyses (Figure 1). Muscles form a compact configuration, occupying a substantial cross-sectional layer within the limb. Contrarily, tendons are short and sturdy. Through dissection and histological examination, the presence of the sesamoid (“ulnar patella”) (Figure 2) has been corroborated. It is a small cartilage nonosseous structure appearing symmetrically in both forelimbs. It is located between the *condylus ulnaris* of the humerus and the *olecranon* of the ulna within the ligament of this joint. It displayed remarkable flexibility while exhibiting a hardened central region characterized by multilayered connective tissue. The periosteal aspect of the structure exhibits a shiny, glassy appearance, presenting a smooth surface. The architecture of the forelimb muscles displays heterogeneity. Certain muscles merge, resulting in fiber fusion, and the number of tendons decreases, ultimately leading to singular insertions. The fascia of compact fibrous connective tissue is reduced, and adipose tissue is predominantly absent beyond the brachial plexus along the path of major vessels and nerves. Nerves and vessels in the limb course proximally to the bone, histologically forming substantial bundles (unlike, e.g., in mammals).

The entire limb functions as a robust support for the well-developed muscles of the neck and back (Figure 3). Among these muscles, the *M. serratus ventralis cervicis* plays a significant role, enabling the lifting of the scapula and providing limb stabilization. It also likely aids in maneuvering and changing the direction of movement by altering the neck’s position. Additionally, the *M. pronator teres major* and the *M. pronator teres minor*, together with other scapular muscles, provide further stabilization. In the musculature of the shoulder blade, the layering of individual parts and the diversification of the course of muscle fibers are noticeable. These muscles primarily perform anti-gravity functions, contributing to postural stability before engaging in swinging and movement. They are much more developed in the caudal part of the scapula. A distinguishing feature of this anatomy is the presence of the *M. coracobrachialis*, a muscle that is less commonly observed in other animal species; it is characterized by having two heads. The *M. coracobrachialis* is relatively short and thick muscle. It connects the limb with the shoulder blade and effectively mitigates tension resulting from the spreading posture. The intrinsic muscles of the scapula, such as the *M. serratus ventralis cervicis* and the *M. subscapularis*, are flat muscles rich in connective tissue and fascia.

The *M. deltoideus* comprises the *pars scapularis et acromialis*. These muscles are highly developed, providing essential support for the limbs’ spreading position and the heavy trunk (Figure 3, Figure 4 and Figure 5). They exhibit the largest mass compared to other limb muscles. Remarkably, extensive muscles are observed in the neck and back regions, including large muscle complexes such as the *M. latissimus dorsi* and the *M. trapezius*. Both muscles display well-defined muscle fibers arranged radially around the limb and have an extensive area of initial attachment. This unique arrangement grants the entire limb a wide range of rotation, with a notable predominance of adduction and abduction, corresponding to the lizard’s characteristic mode of movement in the cranial–caudal direction.

Among the arm muscles, the following have been distinguished: the *M. triceps brachii*, the *M. biceps brachii*, and the *M. brachialis*. In the forearm, muscles include the *M. flexor carpi ulnaris*, the *M. extensor carpi ulnaris*, the *M. extensor carpi radialis*, the *M. extensor digitorum communis*, the *M. flexor digitorum communis*, the *M. flexor carpi radialis*, the *M. extensor carpi ulnaris*, the *M. extensor digitorum longus*, the *M. flexor digitorum communis*, the *M. brachioradialis*, and the *M. extensor digitorum longus* (Figure 6, Figure 7 and Figure 8).

The *M. triceps brachii* consists of three heads (*caput laterale*, *longum* and *mediale*). The longum head is the largest within the complex, distinguished by its broad belly and the highest degree of cohesiveness. During the gait of this lizard, especially while running, a noticeable limb protrusion from the shoulder girdle occurs, followed by the subsequent action of the key triceps and biceps muscles. These two muscles also exhibit the highest development in the entire arm and forearm (Figure 4, Figure 5, Figure 6, Figure 7 and Figure 8). The *M. triceps brachii caput longum* is clearly marked in the limb’s anatomical topography when the lizard moves along the scales. Together with the *M. biceps brachii* and the *M. brachialis*, these muscles manifest the most substantial cross-sectional area and a thicker diameter of muscle fibers, clearly distinct from the forearm muscles. Observations indicate that the forelimb contributes less to propulsion, while a stronger propulsive role is carried out by the hind limbs, which display a more robust swinging motion. The primary function of the pectoral limbs is to provide support for the heavy neck. Together with the head, they sway from left to right and right to left during the walk. The forearm muscles correct the trajectory of movement, stabilizing the gait and ensuring precise navigation. These muscles are also active when the animal lowers and raises its head, monitors the ground with its tongue, feeds, or drinks. The sideways movement of the head is closely linked to the neck’s motion, involving slight adjustments in the limb angle relative to the trunk. The limbs move alternately during walking. When perceiving a threat, the lizard displays defensive behavior by hissing with an open mouth, accompanied by a strong shift in body weight from one limb to the other and a swinging motion of its neck. This activity leads to a noticeable rotation and elevation of the forearms. The forearm muscles are thus sufficiently strong enough to enable a slight forward lift. The *M. biceps brachii* consist of two heads (*caput laterale and mediale*), with the lateral head being larger (Figure 3, Figure 7 and Figure 8).

The *M. pronator teres*, upon contraction, enables the rotation of the forearm and enhances the grip strength, leading to increased precision in activities performed with the hand. Moreover, the *M. pronator quadratus* plays a crucial role in the rotation of the radius bone relative to the ulna. This muscle stabilizes the arrangement of these bones and facilitates precise movements of the wrist while supporting grip strength. It traverses a gap where the bones of the forearm do not make direct contact. As the deepest flexor of the hand (Figure 9), this muscle exhibits a long, thin tendon with visible striped fibers. Rotations of the limb are most noticeable during movement, especially during specific activities—while eating, copulating, climbing trees, and digging.

The *M. flexor carpi ulnaris* plays a role in coordinating wrist and hand movements in the medial direction, allowing for the bending of the entire hand. Additionally, it enables tilting of the wrist towards the fifth finger and provides support for grip strength. These muscles, along with other forearm extensors and flexors, are of paramount importance during the lizard’s arboreal life stage. The flexibility of the wrist in its locomotor functions enables the lizard to move efficiently on terrain with varying slopes. Moreover, during prey capture, the lizard utilizes the strength of its wrists to stabilize itself on the ground. The *M. extensor digitorum longus* allows the animal to straighten and lift its fingers. While the lizard does not frequently use prey holding, it may be employed when moving the prey under the belly. It is normal locomotion and prey-fighting behavior, and the lizard tends to raise its hand towards the elbow. Another significant muscle, the *M. extensor carpi radialis*, supports grip strength and hand movements, controls the extension, and ensures the wrist’s overload strength. On the other hand, the *M. extensor carpi ulnaris* facilitates wrist straightening, abduction, and stabilization whilst also interacting with other forearm muscles. Carpal flexion is also provided by the *M. ulnaris lateralis* (Figure 4, Figure 5, Figure 6, Figure 7 and Figure 8).

The *M. flexor digitorum communis* is responsible for grip strength, manipulation, and flexion of the fingers. The *M. extensor digitorum communis* passes through the *retinaculum extensorum* and extends into the deep muscles of the II, III and IV fingers. These muscles attach to the proximal finger joints. The muscle dedicated to the first finger from the forearm has not been indicated. The wrist has an additional bone related to other varanid lizards (*os pisiforme I* et *II*) (Figure 1 and Figure 9). Finger I consists of two *phalanges*, finger II consists of three *phalanges*, finger III consists of four *phalanges*, finger IV consists of five *phalanges,* and finger V consists of 3 *phalanges* (Figure 1).

The *M. flexor digitorum profundus* allows for flexion of the fingers at the interphalangeal and metacarpophalangeal joints. This allows the lizard to effectively close its hand and grasp objects. This feature can be particularly useful for juveniles that occupy a distinct ecological niche. While these hand adaptations may not be commonly observed, instances were noted in which the lizard dug a hole in the ground to create a nest. In smaller lizards, clear grasping movements of objects were observed while the animal climbs (Figure 9 and Figure 10).

The *Mm. lumbricales* contribute to flexion at the metacarpophalangeal joints. They allow for both finger bending and contribute to phalangeal extension, as well as stabilizing finger muscles of the flexors and extensors (Figure 9). They are involved in precise finger movements. However, in adult individuals, their functional significance may be reduced. Positioned on the sides of the bones, the *Mm. lumbricales* serve to cushion the fingers that come into contact with each other, thereby promoting spreading when moving against resistance, such as in sand or water. Juveniles exhibit a noticeable wide spread of their toes during both walking and climbing. While climbing an oval object, the young lizard adeptly bends its phalanges and uses its sharp claws to grip the surface. The second hand emerges from the egg immediately after the head during the hatching. In juveniles, there is a pronounced, strong limb extension while observing the terrain, with greater support from the arm muscles as the forearm contacts the ground. The movement of young lizards is more agile, and their wrists exhibit greater flexibility, distinguishing them from adults.

Certain similarities can be observed between the flexors and extensors of the forearm and wrist; muscles performing opposing functions exhibit comparable mass.

Beneath the skin, fat is relatively inaccessible. In the distal parts of the fingers, the scales transition into the fascia that envelops the bones. The last row of scales forms a claw fold, turning into a coronary band. The claws are supported by ossified tendons (Figure 11). The hand consists of a total of 27 bones.

The specimen was studied in a wide range of topics. Although this is not the main goal, the interesting architecture of muscle fibers is noteworthy to mention. The mean width of muscle fibers within the forelimb surpassed that of the trunk muscles. These fibers exhibited considerable thickness (76–112 µm) and visible striations (Figure 12 and Figure 13). In their course, potential foci of calcification were identified in two preparations (however, no supplementary tests were undertaken). The standard deviation of measured muscle fiber widths slightly exceeded that of the primary muscle components within the thoracic limb. Simultaneously, crucial locomotive muscles had the greatest mass (Table 1 and Table 2).

## 4. Discussion

The anatomy of the forelimb in *V. komodoensis*’s exhibits distinctive features that are unique to this species. Crocodiles, iguanas and other Varanidae species had the greatest comparative importance in this research. Despite being a normal-limbed squamous lizard, it possesses a specialized forelimb. This limb consists of the scapula, coracoid bone, clavicle, humerus, ulna, radius, metacarpals, and phalanges. The skeletal system of the Komodo dragon’s forelimb has already been well described by other authors [25,50], and no differences have been observed in this individual female specimen. The form of the scapula in this species shows functional innovation. It provides greater support for large muscle groups—which combines a muscular, heavy neck and back in conjunction with a well-developed limb. This aids in sustaining the spread position of the limbs, performing anti-gravity functions, and facilitating locomotion. The scapula occurs together with the coracoid bone (also referred to as scapulocoracoid or coracoscapula) [12,50,71]. A significant role here is attributed to the *M. deltoideus pars scapularis et acromialis* (Figure 6), along with the intrinsic muscles of the scapula. The *M. serratus ventralis cervicis* contributes to maneuvering the direction of movement by adjusting the neck’s position. When contracted, these muscles enable the animal to lift its limbs both upwards and sideways (Figure 6 and Figure 7) [71]. The presence of pain in the neck and shoulder or evident difficulties in directional changes and balance stemming from the cervical region may indicate potential issues in this area. Additionally, the *M. serratus ventralis cervicis*, along with other forearm muscles, are responsible for active movement, although its contribution diminishes during the movement phase [71]. In *V. exanthematicus*, it has been demonstrated that humeral movement involves not only the forward and backward movement but also rotation. Many Squamata species have this complex structure (referred to as “scapulocoracoid”) and are described with coracoid and scapulocoracoid processes. The loss of the coracoid process, as well as the loss of clavicles, occurs, for example, among Chamaeleonidae. The coracoid process is also absent in Gekkonidae and Helodermatidae, along with the loss of the interclavicle lateral arms. Strongly developed coracoids occur in Cordylidae [33,72,73]. In *V. exanthematicus*, functional anatomy studies have revealed that the presence of this specific scapula increases the stride length without extending the range of motion of the shoulder and elbow joints [71]. The *M. coracobrachialis* has two heads, stably attaching the scapula. This not only allows the animal to adapt from an arboreal to a terrestrial habitat, but also responds to the rotation of the ulna and radius in relation to the humerus. This rotation alters the moment arm and the line of action of the muscle, promoting strength development rather than a faster sprint. The specific gait is optimized for movement efficiency (following the economy) [56]. Similar gait patterns have been described in other monitor lizards, e.g., *Varanus salvator*—which results from “30–40% of humeral long axis rotation” and 40–55% humeral protraction-retraction [72]—and *Varanus exanthematicus* [71,74]. Conducting comparative studies of the forelimbs between groups of adults and juveniles would be valuable due to the transition from an arboreal to a strictly terrestrial habitat. It is plausible that the scapula may undergo reorganization, and the synsarcosis could change due to the lower weight and specific needs of the animal in relation to its ecological niche (possibly emphasizing agility, speed or shoulder girdle flexibility). These differences were noted in the juvenile forms of New Zealand geckos, particularly in the hind limb and pelvis structure [75]. The shoulder joint of the Komodo dragon itself has less mobility than, for instance, the swimming *V. salvator* [50]. Arboreal lizards have a prominent *M. flexor carpi radialis*, facilitating stronger wrist flexion that aids in gripping branches and climbing [49]. The gait of juveniles and adults is significantly different, suggesting that the scapula muscles may be more delicate in juveniles, while the forearm and hand muscles are more developed. The enhanced wrist flexibility and lighter gait observed in juveniles allow for exceptionally agile movements, contributing to higher speed. As the animal grows and matures, it may encounter fewer natural predators, potentially leading to a decline in agility and speed. Notably, the examined specimen exhibited an absence of a structured flexor plate (superficial palmar aponeurosis) [76]. This may suggest a potential reduction in grip strength, diminished coordination and precision in finger movement, a decrease in overall stability, and an elevated susceptibility to injuries and discomfort. Therefore, it is advisable to take note of the presence of the flexor plate in comparative studies. In the *M. pronator teres* muscle, confirmation can also be found. It serves an important function in maintaining balance and proportion between the forearm muscles. In animals, this muscle maintains plasticity towards the bone rotation angle, which can increase the strength and efficiency of limb mobility and enable precise movements such as bringing the hand to the mouth [77]. While this function might not be fully utilized by adult lizards, it could be particularly important during the juvenile stage.

We can see peculiar solutions in the locomotor system of *V. komodoensis* of the forelimb associated with ossified structures. The sesamoid of the elbow joint (also described as “ulnar patella” or ulnulae [78] (Figure 11), may indicate an increased biomechanical advantage (an enhanced anti-gravity lever system of the elbow joint); it was already found in the forelimb of many taxa, especially lizards [79,80]. This structure is associated with climbing and digging, which increases the leverage of muscles in the current joint [75] and alleviates stress in the muscle or tendon [81]. The animal has a compact arrangement of muscle tissue, firmly anchored to the bones through thick tendons and more proximal innervation. The animal is capable of adapting to a specific gait and moving the center of gravity in the cephalad direction. This allows them to run very fast despite their gigantic size, although this is not sustained over long distances. The scales, along with the skin and subcutaneous tissue, firmly adhere to the underlying layers, establishing connections with the bone structures within the finger area. The limb is tightly attached to the shoulder girdle. The muscles of the chest and shoulder blades make up a large part of all the muscles of the cranial center of gravity, which explains the peculiarity of the spreading limbs. A notable role is played by the intricate composition of the *M. triceps brachii* (comprising three heads) [69,70,71] as well as the *M. biceps brachii* (which possesses two heads). The lateral head, as a significant component of the muscle structure, plays a crucial role in enhancing muscle cohesion, similar to supporting weight and extending the limb outward.

The hindlimb is 6 cm longer than the forelimb, and while the forearm and lower leg share analogies in the location of individual muscles, these differences significantly impact locomotion. The need for separate analyzes of the forelimb and hindlimb was also emphasized by Padian K. and Olsen E. They invoked some analogies of gait and limb positioning to those of crocodiles. A Komodo dragon’s footprint bears resemblance to prints found in “Triassic pseudosuchian theodonts and Early Jurassic crocodiles” [82]. Most correspondences of the muscles of the thoracic limb were observed with crocodiles, other monitor lizards, and iguanas. The hindlimb is characterized by a stronger swing, whereas the forelimb takes a more prominent role in navigation, correcting movement direction, and executing specific complex behaviors such as hunting, digging, copulating, and hatching. The presented research sheds new light on the directions of studies concerning the musculature of Komodo dragon limbs, aiming to better understand its locomotor capabilities. Observations and knowledge of the anatomy of the key limb muscles suggest that lameness may manifest as changes in neck and head movement. Disproportionate development of the thoracic limb may lead to unilateral bending of the neck, tilting of the head, and difficulty maintaining a straight gait. The key arm muscles exhibit a greater cross-sectional width of muscle fibers and have the capacity to grow faster during growth and development. In juveniles, a more agile gait and a more prehensile limb are advantageous. However, as the animal grows to gigantic proportions, muscle development tends to prioritize anti-gravity abilities and the capacity to capture large-sized prey. Consequently, the arm muscles play a crucial role in limb propulsion and repulsion. The opposing sides of the limb display similarly developed muscle mass, although variations in the size between flexors or extensors do not always correlate with a cross-sectional size of muscle fibers. The forearm shows more diverse muscle courses and performs a wider range of movements despite these specialized functions not being directly related to gait. Various specializations can be observed, providing predispositions for specific movements in the shoulder girdle, arm, forearm, and wrist. During gait, running, kicking, copulation, and hatching, the final effect of the body movement involves setting rotated, spreading limbs relative to each other. The animal employs a specific set of complex movements to skillfully transfer the weight force and maximize muscle strength in key areas. The limb movement harmonizes with the extensive muscles of the neck and back, leading to noticeable symmetry in the movement of the front and back parts of the body, as well as the right and left sides. Disproportions in the construction of these key units, whether in relation to the right and left side or the fore and hind limbs, may indicate developmental disorders or incorrect living conditions of the animal. These discrepancies can be assessed during postmortem examinations, but this requires performing biometric tests on a larger number of individuals, both adult and young. Similar relationships have already been demonstrated in the construction and shape of the claws.

The limbs are equipped with large, sharp claws, which are supported by ossified tendons. Their function in relation to hurting prey is rather less significant, but more anchoring in the ground and being able to pull up the whole body whilst eating, as well as removing sticking elements from the mouth [59] and in locomotion. Claws reflect some specialization related to body size, habitat and selection behaviors, showing variability of structure, especially shape and length. In this, it is assumed that the climbing animal has a more strongly angulated claw, and as its body mass increases in a terrestrial environment, the claws will be longer and less crooked [83]. In captivity, it happens that the claws overgrow and require regular trimming. Foot callous lesions and wounds are also common (especially in overweight individuals or with arthritis) [22,84]. The Komodo dragons live in a savannah-type landscape with an angle of inclination from 10 up to 40 degrees of the habitat [20]. They may also be predisposed to build up some key muscles in limbs, and this allows them to keep parts of the growing claws properly filed, preventing lameness and injuries. It seems that the hand muscles in the adult are much less pronounced than they might be in juveniles. However, the final point of body weight transfer during movement is the hand itself. It is characterized by additional support from ossified tendons, more fascia and connective tissue, and thicker skin, along with the formation of thickened, calloused skin in the place of fingertips. These bear a variety of scales, which are small, sharply pointed and triangular-shaped in appearance.

The connection of this information about the Komodo dragon gives new insights into its morphology. The presence of strongly developed specific muscles that strengthen the shoulder blade, along with the support provided by sesamoid bones and ossified tendons, contributes to a specialized hand with greater flexibility. All these aspects indicate the potential differentiation of the limb structure at different stages of this lizard’s life, as well as its adaptation to a typically terrestrial habitat. This adaptation serves as a solution to the challenge of gigantism, achieved through the architectural distribution of stress and tension associated with gravity and forces during the animal’s movements. Furthermore, the inconspicuous reserve of subcutaneous fat and the firm, almost fused scales containing muscle fragments show the key role that muscles play in protecting the skeleton.

Understanding the biology of the Komodo dragon holds significant importance for the conservation of this species and for providing it with the best conditions in zoos. This unique species may give us an insight into the reasons behind the extinction of many of the world’s megafauna. It is a way to explore evolutionary biomechanics, particularly in relation to terrestrial giants [85]. The conducted research offers valuable insight into anatomy and physiology, and the collected materials may be valuable for veterinarians. The findings emphasize the significance of limb angulation specificity, muscle mass proportion, proper nutrition, and the natural behavior of this reptile. Assessing the specific gait and taking measurements of the thoracic limb can give a lot of information about the quality of life of a lizard.

## 5. Conclusions

The Komodo dragon has individual features of the anatomy in its thoracic limb muscles, distinguishing it from other lizards. Strongly developed muscle groups are the result of transferring the body weight to the head and maintaining the spread position of the limbs. It is an animal that can provide a better insight into evolutionary changes and an explanation of intermediate forms in the field of anatomy, physiology, and behavior. It is an endemic animal whose environmental optimum is in the place of residence. However, these observations already indicate that limiting the biodiversity of the Indonesian environment, reducing the food pool, and extending the distance to its acquisition may have a real negative impact on this species. These lizards are able to run fast but only for a short distance. They have species-specific adaptations of musculoskeletal structures that give a better insight into the biology of the species and can be developed by other researchers and be inspiring for biorobotics. By making the anatomical study available, the potential for significant knowledge gain extends beyond this species. However, one should remember about the diversity arising from factors such as territory, sex and age. Extensive comparative research can be conducted on this basis.

## Figures and Tables

**Figure 1 animals-13-02895-f001:**
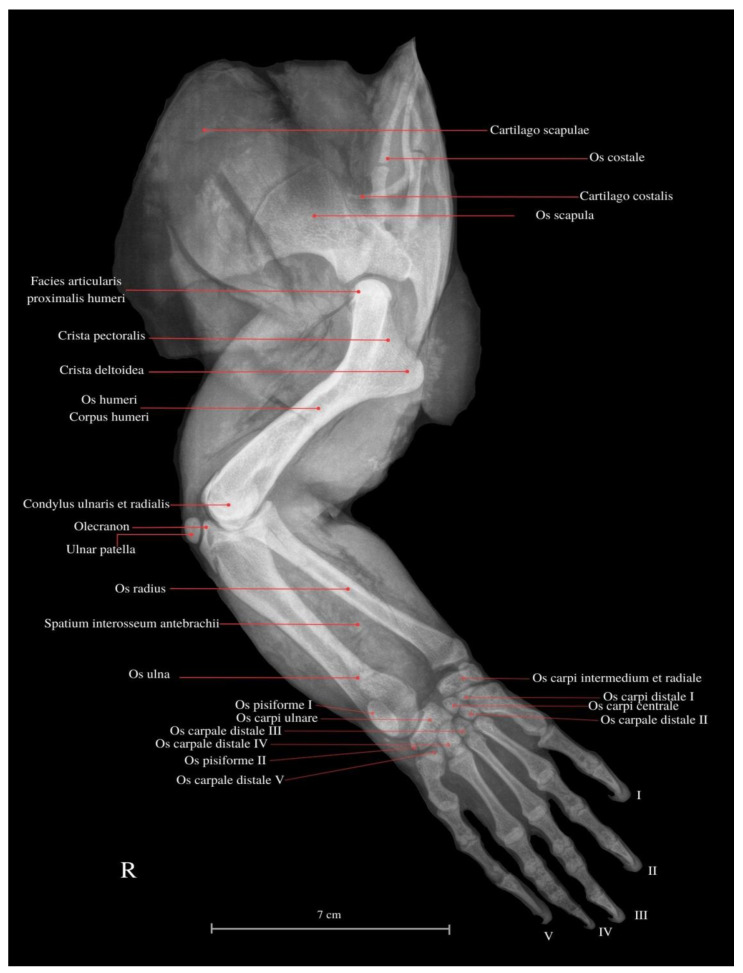
Lateral view of the thoracic limb of the Komodo dragon (X-rays). (R) Right side. *Cartilago scapulae*—scapular cartilage; *Os costale*—rib bone; *Cartilago costalis*—costal cartilage; *Os scapula*—shoulder blade bone; *Facies articularis proximalis humeri*—proximal humeral joint surface; *Crista pectoralis*—pectoral crest; *Crista deltoidea*—deltoid crest; *Os humeri*—shoulder bone; *Corpus humeri*—humerus shaft; *Condylus ulnaris et radius*—ulnar and radial condyles; *Olecranon*—ulnar process; *Ulnar patella*—patella-like ulna, sesamoid; *Os radius*—radius bone; *Spatium interosseum antebrachii*—interosseal space of the forearm; *Os ulna*—ulnar bone; *Os pisiforme I*—pisiform bone I; *Os pisiforme II*—pisiform bone II; *Os carpi ulnare*—ulnar carpal bone; *Os carpi distale I*—distal wrist bone I; *Os carpi distale II*—distal wrist bone II; *Os carpale distale III*—distal wrist bone III; *Os carpi intermedium et radiale*—intermediate and radial wrist bones; *Os carpi centrale*—central wrist bone; *Os carpale distale V*—distal wrist bone V. I—finger I; II—finger II; III—finger III; IV—finger IV; V—finger V.

**Figure 2 animals-13-02895-f002:**
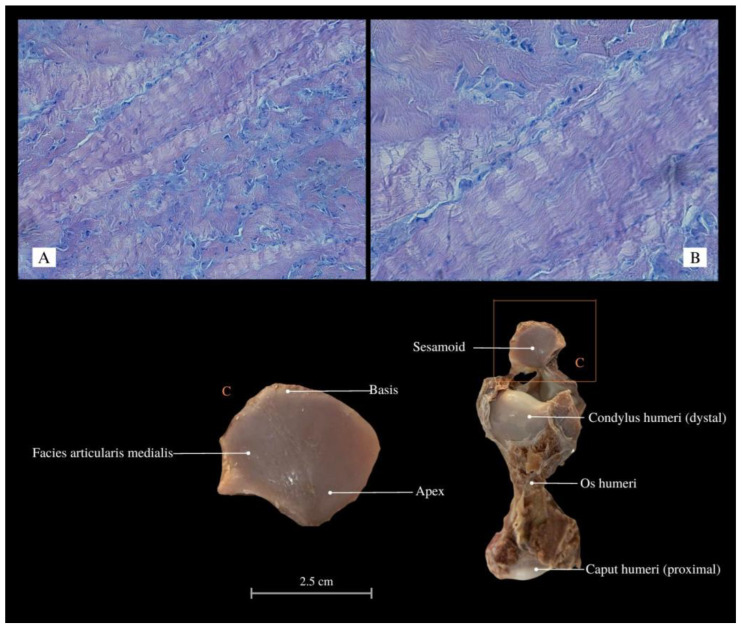
Histological images (HE stained) of the sesamoid (elastic cartilage, “ulnar patella”) with chondrocytes and chondroblasts arranged in bands between collagen fibers, Mag 200× (**A**); Mag 400× (**B**). Macroanatomical views of the sesamoid (**C**). *Apex*—top; *Basis*—base; *Facies articularis medialis*—medial articular facet; *Condylus humeri*—condyle of humerus; *Os humeri*—humerus bone; *Caput humeri*—head of humerus.

**Figure 3 animals-13-02895-f003:**
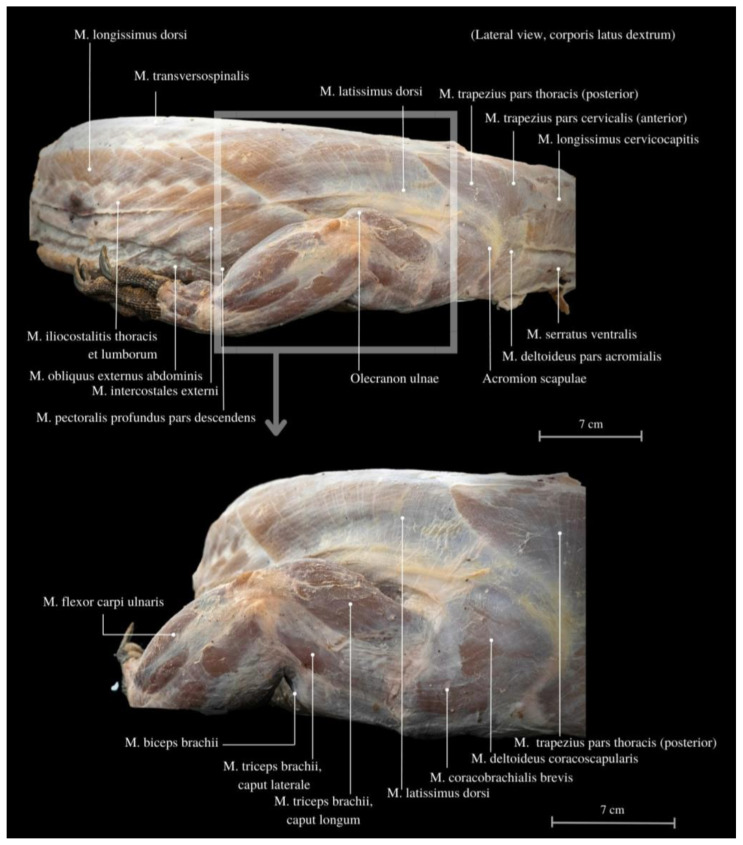
Muscles of the pectoral girdle muscles, along with the thoracic and abdominal structures. *M. longissimus dorsi*—longissimus muscle; *M. transversospinalis*—transversospinal muscle; *M. latissimus dorsi*—the lats; *M. trapezius pars thoracis (posterior)*—trapezius muscle, thoracic part (posterior); *M. trapezius pars cervicalis (anterior)*—trapezius muscle, cervical region (anterior); *M. longissimus cervicocapitis*—the longest cervicocapitis muscle; *M. serratus ventralis*—serrated ventral muscle; *M. deltoideus pars acromialis*—deltoid muscle, acromial part; *Acromion scapulae*—cromion of the scapula; *Olecranon ulnae*—ulnar process; *M. pectoralis profundus pars descendens*—the deep pectoral muscle, descending part; *M. intercostales externi*—external intercostal muscles; *M. obliquus externus abdominis*—external oblique muscle of the abdomen; *M. iliocostalitis thoracis et lumborum*—iliocostalis muscle of the thoracic and lumbar regions; *M. flexor carpi ulnaris*—flexor carpi ulnaris muscle; *M. biceps brachii*—biceps brachii muscle; *M. triceps brachii, caput laterale*—triceps brachii muscle, lateral head; *M. triceps brachii, caput longum*—triceps brachii muscle, long head; *M. coracobrachialis brevis*—short coracobrachialis muscle; *M. deltoideus coracoscapularis*—deltoid muscle, coracoscapular part; *M. trapezius pars thoracis (posterior)*—trapezius muscle, thoracic part (posterior).

**Figure 4 animals-13-02895-f004:**
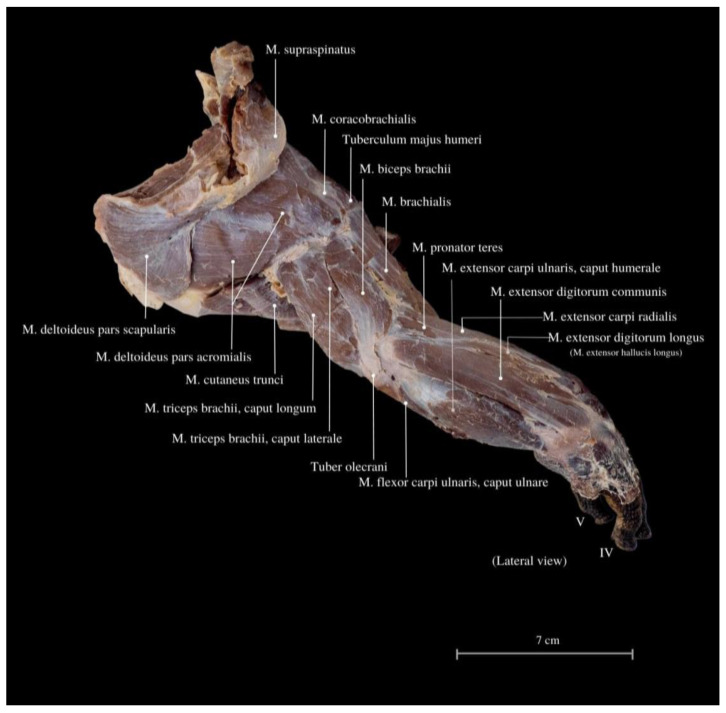
Lateral view of the muscles of the thoracic limb. *M. supraspinatus*—supraspinal muscle; *M. coracobrachialis*—coracobrachial muscle; *Tuberculum majus humeri*—greater tubercle of the humerus; *M. biceps brachii*—biceps brachii muscle; *M. brachialis*—brachialis muscle; *M. pronator teres*—round pronator muscle; *M. extensor carpi ulnaris, caput humerale*—extensor carpi ulnaris muscle, humeral head; *M. extensor digitorum communis*—extensor digitorum—common extensor muscle; *M. extensor carpi radialis*—extensor carpi radialis muscle; *M. extensor digitorum longus (M. extensor hallucis longus)*—fingers long extensor muscle; *M. deltoideus pars scapularis*—deltoid muscle, scapular part; *M. deltoideus pars acromialis*—deltoid muscle, acromial part; *M. cutaneus trunci*—cutaneous trunk muscle; *M. triceps brachii, caput longum*—triceps arm muscle, long head; *M. triceps brachii, caput laterale*—triceps arm muscle, lateral head; *Tuber olecrani*—tuber of the ulnar process; *M. flexor carpi ulnaris, caput ulnare*—flexor carpi ulnar muscle, ulnar head. IV—finger IV; V—finger V.

**Figure 5 animals-13-02895-f005:**
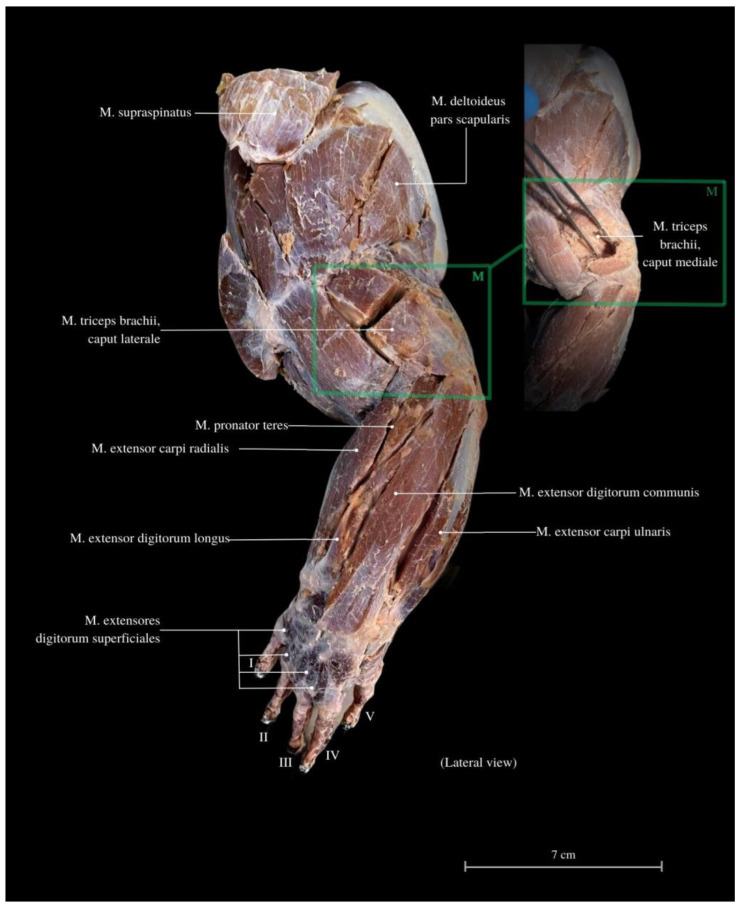
Lateral view of the thoracic limb. *M. supraspinatus*—supraspinal muscle; *M. triceps brachii, caput laterale*—triceps arm muscle, lateral head; *M. triceps brachii, caput mediale*—triceps arm muscle, medial head; *M. pronator teres*—round pronator muscle; *M. extensor carpi radialis*—extensor carpi radialis muscle; *M. extensor digitorum longus*—fingers long extensor muscle; *M. extensor digitorum superficialis*—fingers superficial extensor muscle; *M. extensor digitorum communis*—commonous fingers extensor muscle; *M. extensor carpi ulnaris*—extensor carpi ulnaris muscle; *M. deltoideus pars acromialis*—deltoid muscle, acromial part. I—finger I; II—finger II; III—finger III; IV—finger IV, V—finger V.

**Figure 6 animals-13-02895-f006:**
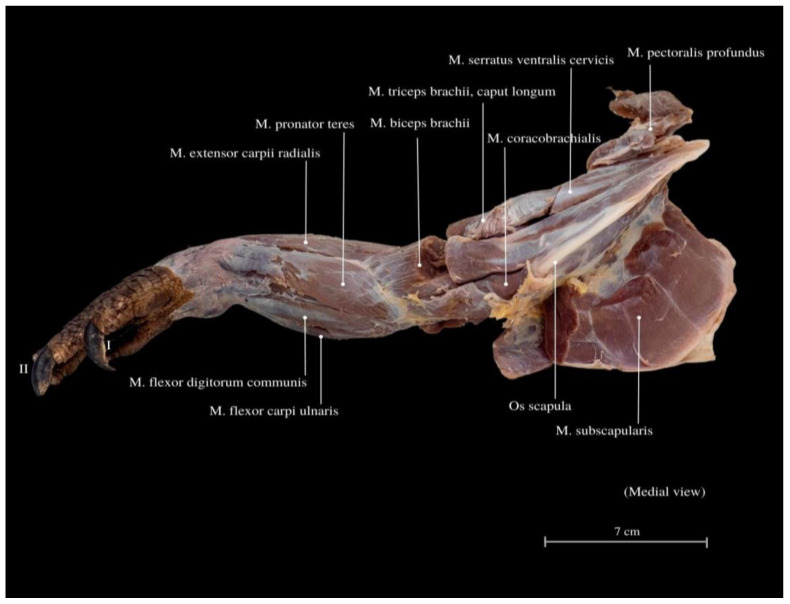
Medial view of the muscles in the thoracic limb. *M. pectoralis profundus*—the deep pectoral muscle; *M. serratus ventralis cervicis*—serratus ventral cervical muscle; *M. biceps brachii*—biceps brachii muscle; *M. extensor carpi radialis*—extensor carpi radialis muscle; *M. flexor digitorum communis*—commonous fingers bending muscle; *Os scapula*—shoulder bone; *M. subscapularis*—subscapular muscle. I—finger I; II—finger II.

**Figure 7 animals-13-02895-f007:**
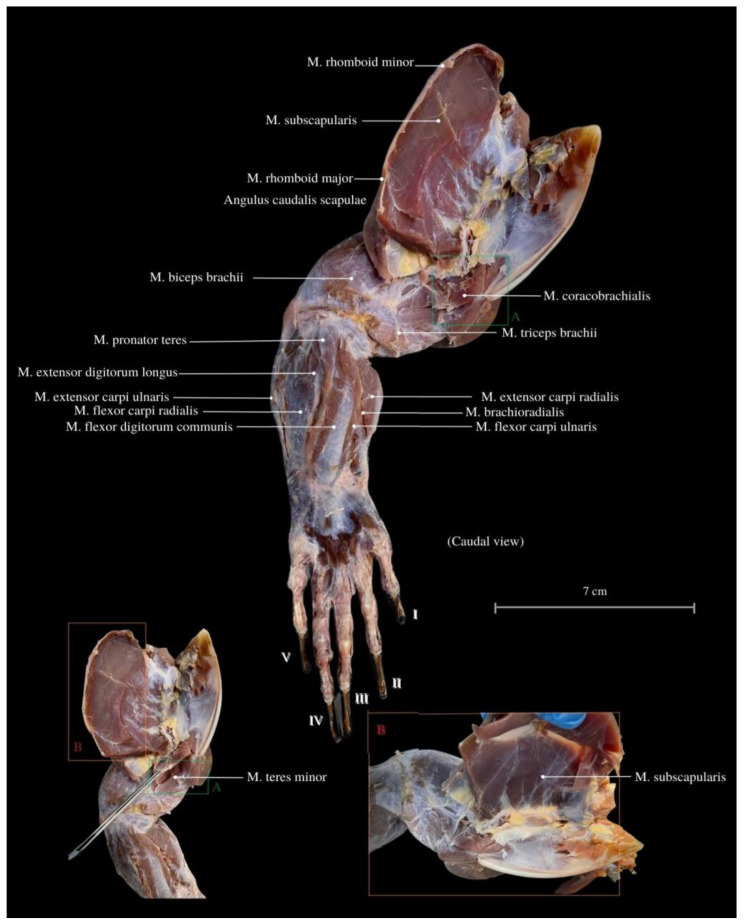
Caudal view of the thoracic limb. *M. subscapularis*—subscapular muscle; *M. rhomboid major*—rhomboid major muscle; *M. rhomboid minor*—rhomboideus smaller muscle; *Angulus caudalis scapulae*—caudal angle of the scapula; *M. biceps brachii*—biceps brachii muscle; *M. pronator teres*—round pronator muscle; *M. teres minor* –round smaller muscle; *M. extensor carpi ulnaris*—extensor carpi ulnaris muscle; *M. flexor digitorum communis*—commonous fingers bending muscle; *M. extensor carpi radialis*—extensor carpi radialis muscle; *M. triceps brachii*—triceps arm muscle; *M. flexor carpi ulnaris*—flexor carpi ulnar muscle, ulnar head; *M. brachioradialis*—brachioradial muscle; *M. coracobrachialis*—coracobrachial muscle. I—finger I; II—finger II; III—finger III; IV—finger IV, V—finger V.

**Figure 8 animals-13-02895-f008:**
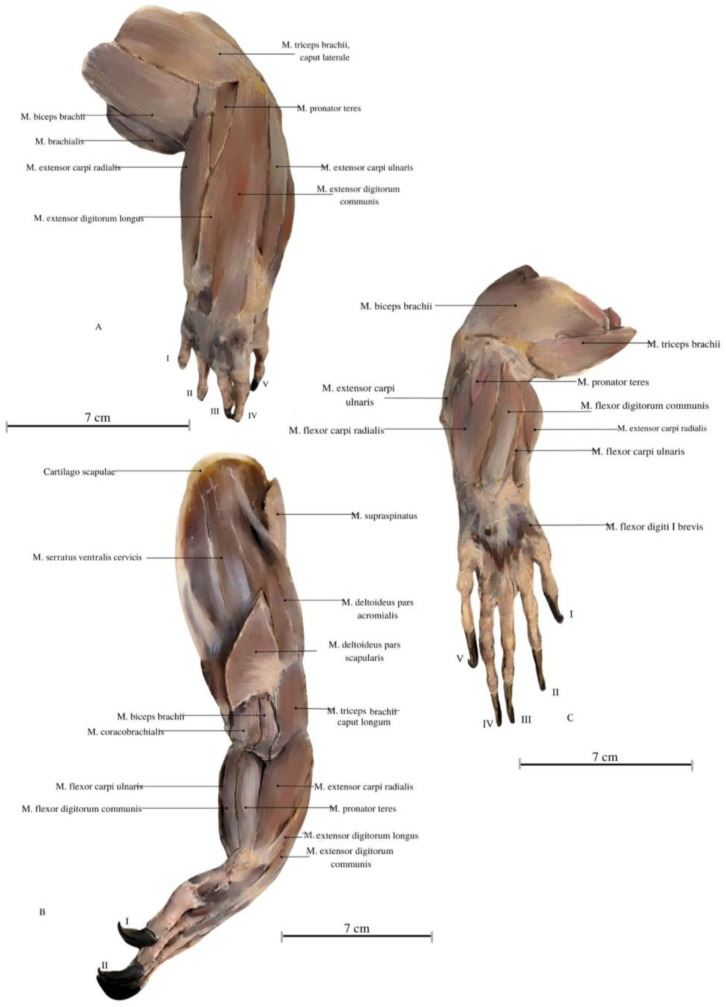
Illustrations of the Komodo dragon’s forelimb muscles. (**A**) Lateral view. (**B**) Oblique view. (**C**) Medial view. Author: A. Tomańska. *M. biceps brachii*—biceps brachii muscle; *M. brachialis*—brachialis muscle; *M. pronator teres*—round pronator muscle; *M. teres minor* –round smaller muscle; *M. extensor carpi ulnaris*—extensor carpi ulnaris muscle; *M. flexor digitorum communis*—commonous fingers bending muscle; *M. extensor carpi radialis*—extensor carpi radialis muscle; *M. extensor carpi ulnaris*—extensor carpi ulnaris muscle; *M. extensor digitorum longus*—fingers long extensor muscle; *M. triceps brachii*—triceps arm muscle; *M. flexor carpi ulnaris*—flexor carpi ulnar muscle, ulnar head; *M. extensor digitorum communis*—commonous fingers extensor muscle; *M. supraspinatus*—supraspinal muscle; *M. brachioradialis*—brachioradial muscle; *Cartilago scapulae*—scapula’s cartilage; *M. serratus ventralis cervicis*—serratus ventral cervical muscle; *M. coracobrachialis*—coracobrachial muscle; *M. deltoideus pars acromialis*—deltoid muscle, acromial part; *M.* flexor digiti I brevis—finger I bending deep muscle. I—finger I; II—finger II; III—finger III; IV—finger IV, V—finger V.

**Figure 9 animals-13-02895-f009:**
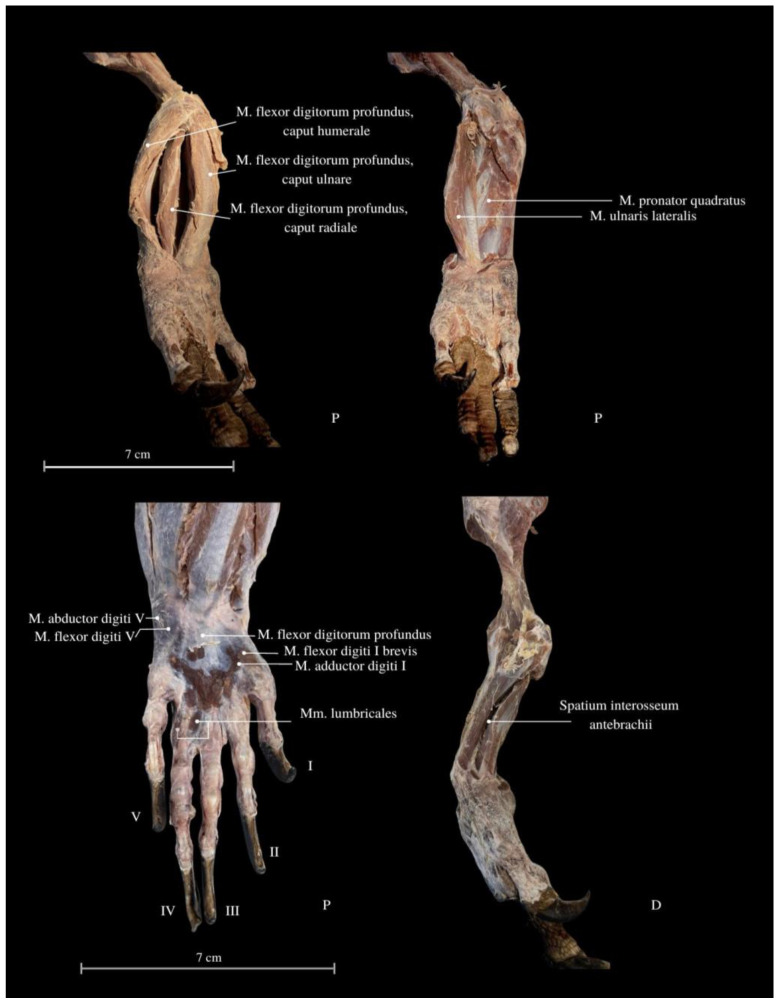
Dorsal (D) and palmar (P) view of the forearm and the carpus. *M. flexor digitorum profundus, caput humerale et ulnare et radiale*—finger deep flexor muscle, humeral, and ulnar, and radial head; *M. pronator quadratus*—pronator quadrates muscle; *M. ulnaris lateralis*—ulnar lateral muscle; *M. abductor digiti V*—approaching finger V muscle; *M. flexor digiti V*—flexor finger V muscle; *M. flexor digitorum profundus*—deep flexor finger muscle; *M. flexor digiti I brevis*—flexor finger I short muscle; *M.* adductor digiti I—leading to finger I muscle; *Mm. lumbricales*—lumbrical muscles; *Spatium interosseum antebrachii*—interosseus space of the forearm. I—finger I; II—finger II; III—finger III; IV—finger IV, V—finger V.

**Figure 10 animals-13-02895-f010:**
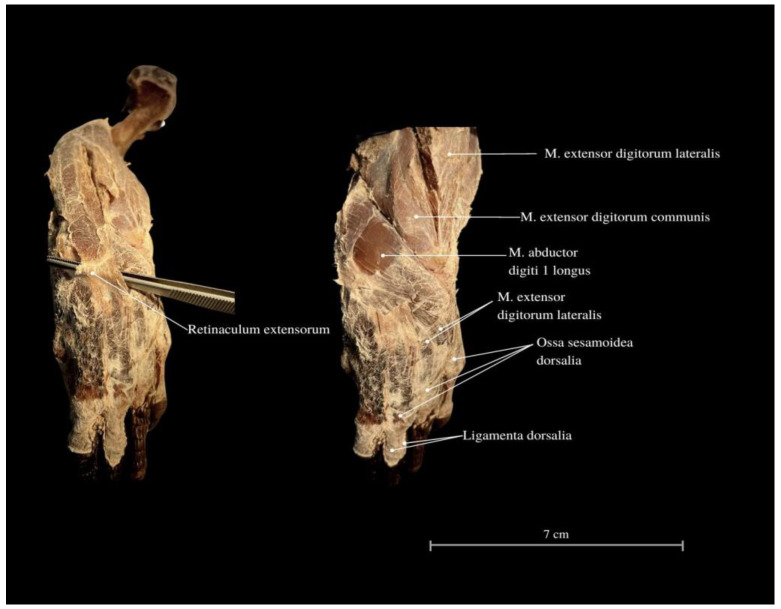
Cranial view of the forearm. *Retinaculum extensorum*—extensor of the retinaculum; *M. extensor digitorum lateralis*—extensor finger lateral muscle; *M. extensor digitorum lateralis*—extensor finger lateral muscle; *M. extensor digitorum communis*—commonous fingers extensor muscle; *Ossa sesamoidea dorsalia*—dorsal sesamoid bones; *Ligamenta dorsalia*—dorsal ligaments.

**Figure 11 animals-13-02895-f011:**
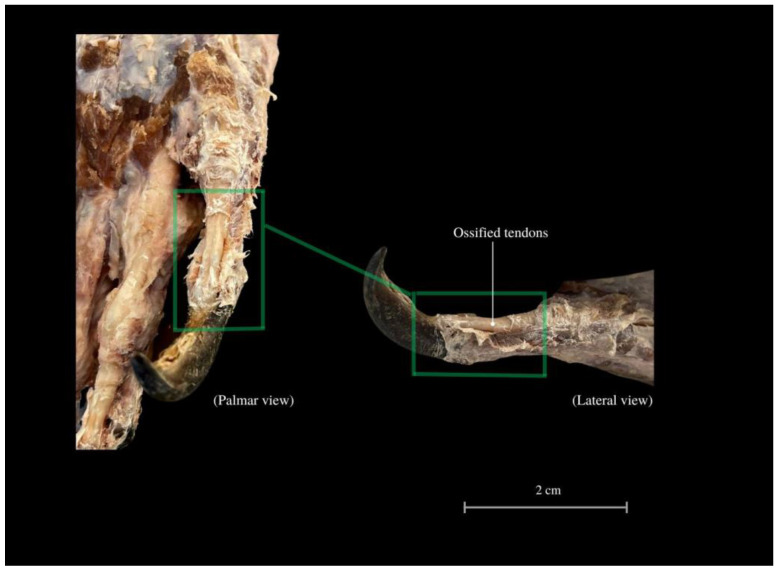
Ossified tendons providing support for the fingers and stability of the claws. *Tendines ossificati*—ossified tendons.

**Figure 12 animals-13-02895-f012:**
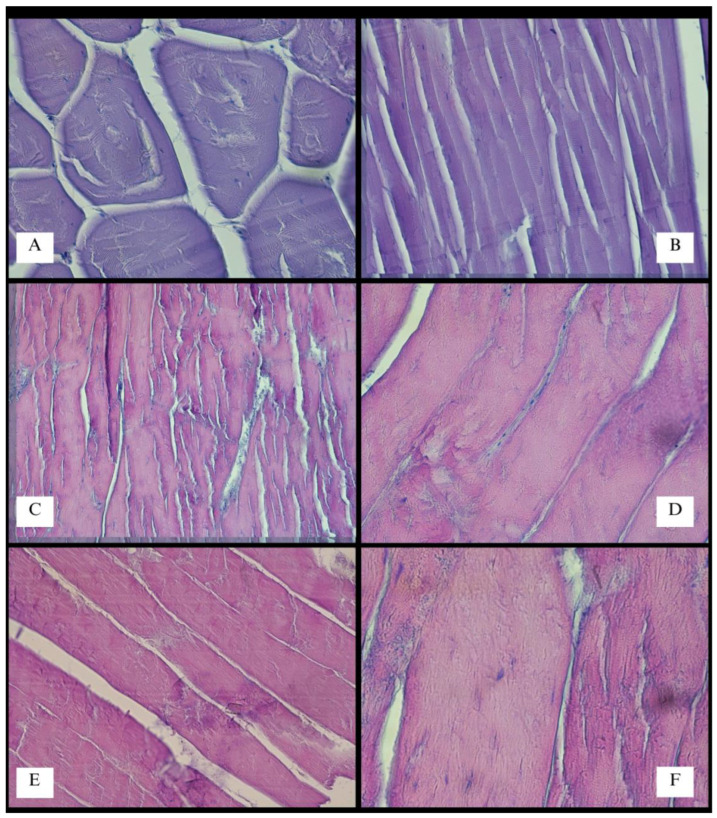
Histology (HE stained) of the selected muscles of *V. komodoensis*. The *M. deltoideus* fibers in cross section, Mag 100× (**A**); the *M. deltoideus*, Mag 200× (**B**); the *M. latissimus dorsi*, lense magnification ×20 (**C**); the *M. latissimus dorsi*, Mag 400× (**D**); the *M. triceps brachii*, Mag 200× (**E**); the *M. latissimus dorsi*, Mag 600× (**F**).

**Figure 13 animals-13-02895-f013:**
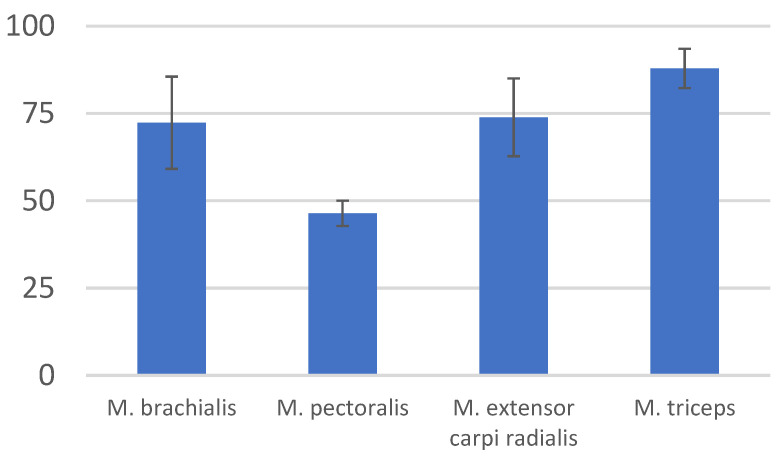
Selected morphometric measurements of muscle fibers with an indication of the average width (µm) and (±SD) standard deviation (Microsoft Corporation, 2018. *Microsoft Excel*, Washington, DC, USA). Each sample was based on 100 morphometric measurements.

**Table 1 animals-13-02895-t001:** Weight of the selected muscles of the thoracic limb of *V. komodoensis*, 30 kg female.

Muscle	Weight (W, g)	W_muscle_/W_body_ (%)
*M. trapezius*	12.02	0.04
*M. deltoideus pars scapularis*	36.44	0.121
*M. deltoideus pars acromialis*	38.87	0.129
*M. triceps brachii, caput longum*	16.37	0.055
*M. triceps brachii, caput mediale*	13.87	0.046
*M. triceps brachii, caput laterale*	10.16	0.034
*M. biceps brachii, caput laterale*	24.15	0.081
*M. biceps brachii, caput mediale*	11.73	0.039
*M. brachialis*	18.27	0.061
*M. pronator teres*	5.08	0.017
*M. extensor digitorum communis superficialis*	11.32	0.038
*M. extensor carpi radialis*	13.45	0.045
*M. extensor carpi ulnaris*	17.83	0.059
*M. extensor digitorum longus (hallucis longus)*	11.88	0.039
*M. flexor digitorum communis superficialis*	13.21	0.044
*M. brachioradialis*	9.65	0.032
*M. flexor carpi ulnaris*	17.83	0.059
*M. flexor carpi radialis*	12.81	0.043
*M. flexor digitorum profundus, caput humerale*	15.7	0.052
*M. flexor digitorum profundus, caput radiale*	14.95	0.049
*M. flexor digitorum profundus, caput ulnare*	13.53	0.045
*M. pronator quadratus*	5.06	0.017
*M. ulnaris lateralis*	8.89	0.029
*M. abductor digiti I longus*	2.53	0.008

**Table 2 animals-13-02895-t002:** Selected morphometric measurements of muscle fibers with an indication of the average width (µm) and (±SD) standard deviation.

Muscle	Mean	Standard Deviation (±SD)
*M. brachialis*	72.35	13.2
*M. pectoralis*	46.41	3.635
*M. extensor carpi radialis*	73.89	11.115
*M. biceps brachii*	87.9	5.65

## Data Availability

No other details regarding where the data supporting reported results can be found, including links to publicly archived datasets analyzed or generated during the study.

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
