# Peer review of "Functional Anatomy of the Thoracic Limb of the Komodo Dragon (Varanus komodoensis)"

_animals, 2023, doi:10.3390/ani13182895_

Round 1

Reviewer 1 Report (Previous Reviewer 1)

I think that Title should be change to "Functional anatomy...." because results are basically talking about functionality of the forelimb and a few about anatomy, This is my opinion after reading several times the results. For example, you have discovered that m. biceps brachii has two heads when in the most domestic and wild mammals have been reported only one head and you do not report it, so something it is wrong.

Author Response

Reviewer 2 Report (Previous Reviewer 2)

Dear authors,

Your manuscript is interesting and much better than its previous version. The results are clearly presented, as is the discussion. The article still requires minor corrections.

First, the language still needs polishing.

lines 136-139 - this sentence is unclear.

line 145 - Trimeresus should be italicized

The abbreviation for musculus (m.) is capitalized in some places and not in others (e.g. lines 308, 314 and others)

line 528 - lunulae or ulnulae?

The Diplometopon/Diplomethoyjon issue you address in response letter - The correct is Diplometopon. The second name is result of error when digitizing paper version into pdf. 

Round 2

Reviewer 1 Report (Previous Reviewer 1)

There are a little mistakes and errors. Please fix it. Figures must be removed from the Materials and methods.

Author Response

This manuscript is a resubmission of an earlier submission. The following is a list of the peer review reports and author responses from that submission.

Round 1

Reviewer 1 Report

Images and dissections must be improved. The anatomical study need help of an veterinary anatomist not a physiologist. Some anatomical terms are not correct. Discussion Komodo Dragon forelimb anatomy must be compared  with other domestic mammals if it is not possible with other lizards dissections.

Reviewer 2 Report

Dear Authors,

I read your paper carefully. You rise interesting and important problem. Anatomy of most recent animals is still poorly- or completely unstudied, and your paper aims to fill one of the gaps. However, at this moment I cannot recommend your paper for publication in any serious journal. Below I provide general remarks for your paper following more specific comments for subsequent sections. I understand that some of my comments may be harsh and bitter, however some problems mentioned below should not appear in academic writing.

Finally, I noticed that the first and the corresponding authors are both students and probably unexperienced researchers and scientific-paper writers. However, their older, more experienced colleagues should put more attention on the manuscript draft and help to improve the paper they read and accepted. I strongly advice 1998 issue of Herpetologica (Vol. 54, Supplement: Points of View on Contemporary Education in Herpetology, and in particular Toft and Jagger papers:

Toft, C. A., & Jaeger, R. G. (1998). Writing for Scientific Journals I: The Manuscript. Herpetologica, 54, S42–S54. http://www.jstor.org/stable/3893287

Jaeger, R. G., & Toft, C. A. (1998). Writing for Scientific Journals II: The Review Process. Herpetologica, 54, S54–S63. http://www.jstor.org/stable/3893288

General remarks

Because of the mess with the References I cannot fully evaluate the paper. It is very hard to follow authors, when I want to check the source of given information and find completely irrelevant paper under the given number, whereas probably right reference is hidden under +1, +2 or +3 number to those referred (see References remarks). This must be fixed.

Except of several editorial problems, I find the paper need extensive reworking, prior any publication attempt. I find your work potentially valuable, but the data are poorly presented and discussion is often irrelevant.

You missed several important papers that should be cited here. I did short google search and have found several important paper on the anatomy of varanid lizards you do not cite (plus some general reviews on squamate myology). This weakens your paper value, since there is lack of theoretical background in the introduction and lack of relevant discussion. I list the missing papers at the end of my review comments. However, it was your homework to be done prior starting dissection and writing. Good start would be: Russell A. P., Bauer A. M. (2008). The appendicular locomotor apparatus of Sphenodon and normal-limbed Squamates. In: Gans C., Gaunt A. S., Adler K. (eds.) Biology of the Reptilia, vol. 21, Morphology I, The Skull and Appendicular Locomotor Apparatus of Lepidosauria. Society for the Study of Amphibians and Reptiles, pp: 1-466.

Introduction

Your paper is dedicated to comparative and functional anatomy. There is no need to discuss the conservation issues of the species (or perhaps single sentence to stress the importance of you study-species), since it brings nothing to the core topic of your paper. Instead I would like to read short review on what do we know about variation of the forelimb musculature in Anguimorpha, and Varanidae in detail; about functional meaning of the muscle arrangement in relation to habitat and body size.

Line 50: Use formal taxon name when first mention: Varanus komodoensis Ouwens, 1912

Lines 55-56: “among which 8 subpopulations [8] have been distinguished in nature, and individuals kept in zoos.” – please delete “and individuals kept in zoos.”

Lines 56-57: “The body size of the komodo dragon hasn’t changed in the last 900,000 years.” – please provide reference for this statement. It is at least controversial, body size can vary between populations.

Lines 57-58: “It is a species that represents gigantism in the animal world, determined on the basis of morphological and genetic studies, constituting a synapomorphy of the clade.” This does not make sense. What is determined on the basis of morphological and genetic studies? Gigantism? What constitutes synapomorphy of the clade? Gigantism? And why to put synapomorphies here, if there is no discussion on the phylogeny?

Lines 59-76: Delete. It is irrelevant in the context of the paper. Stories about lizard predicted extinction have nothing to do with their forearm anatomy.

Lines 77-84: This is interesting!

Lines 102-103: “Reorganisations within the forelimb are documented as evolutionary, the greatest difference has been shown from sprawling animals (specific to monitor lizards) to parasagittal forelimbs (e.g. in mammals).” Discussion on the evolution of mammals is irrelevant. You do not have broad taxonomical sampling and do not conduct study in the phylogenetic framework to discuss such transformations. Varanids and mammals are very, very, very distantly related. The same apply for crocodiles. In general, the section “Active and passive movement system” is very chaotic and poorly written. Lot of information is inadequate. Please focus on what do we know about (1) V. komodoensis, and (2) varanids sensu lato, and perhaps (3) other squamates. After lecture of this section reader should know what species has been already studied, what do other researches found and why do you conduct your study if there is anything more than descriptive purposes. This is lacking.

Line 123: Malayan water monitor, please add formal taxon name: Varanus salvator (Laurenti, 1768)

Lines 145-146: (Alligator mississippiensis Daudin 1801), change to (Alligator mississippiensis [Daudin, 1801])

Please note, that when referring to formal taxon name, when you give both latin name and the name of author of the first description, the comma after author name is mandatory, as it is the rule of the Codex of Zoological Nomenclature. More, the decision if author name is in parentheses or not, also results from the Codex. If author has placed the new taxon in different genus than recently recognized (it is matter of phylogeny and monophyletic grouping), then her/his name is taken in parentheses. In this case, the gator was described by Daudin as Crocodilus mississippiensis, and later was revised to be member of Alligator (A.M.C. Duméril & Bibron, 1836 as A. lucius [Cuvier, 1807] and synonymized with C. mississippiensis by Hoolbrock in 1836 as A. mississippiensis [Daudin, 1801]). Other example of the rule: Ouwens was correct in placing V. komodensis in the genus Varanus, whereas Laurenti was wrong when V. salvator was originally placed by him in the genus Stellio.

Methods

How many specimens? Single one? What was it size? Except of weight and age it is good to know the SVL (Snout to Vent Length; the standard measure of the squamate length).

Lines 222-223: “research on the structure of lizards [38]” – something is missing here…

Lines 220-238: this is the only part that fits this section. However, it lacks some information. Table 1 should be moved to Results section.

Line 230: “thus”? Shouldn’t be “then”?;

Archeozoology Laboratory and Nature Museum – which institution do you mean? Please provide full details (city, country, eventually University to which the Lab and Museum are affiliated etc.);

“compared with the skeleton” – what skeleton? Other V. komodoensis? Other species? Please clarify and provide voucher numbers or other sources (publications you refer to).

Please consider rearrange this section in the following way: start with description of your sample (e.g. We studied single specimen (adult female) of V. komodoensis from the collection of… (voucher number). The specimen was analysed using … (here provide methods of dissection, X-raying etc.). We also used comparative material: list the species number of specimens and their voucher numbers, eventually cite the literature sources.

Results

I find this section poorly organized. Please consider rearrangement of the structure in more systematic way with description of muscle by muscle, each in separate paragraph. Starting with shoulder gridle musculature, stylopodium musculature, zeugopodium musculature and finally autopodium musculature. Good example of such practice is Russell and Bauer (2008). At present form it is hard to extract relevant information. Discuss the results in Discussion section – what are differences and similarities among closely related species, specially members of Varanidae.

Lines 250-255 belong to Discussion rather.

Line 265: Reference to “fig. 11” appears here, but there is no reference to figures 8, 9 and 10. Please arrange the figures in the order they are mentioned in the text.

Lines 285-287: This sentence is unnecessary. Please delete or move it to Introduction and expand the thought.

Table 1. Was the female exactly 30 kg weight? Or is it approximation?

Figure 2. This is nice photo of skeleton of varanid lizard, but of little if any value for comparative anatomy studies. Please provide detail drawings of forelimb skeleton and pectoral girdle. The rest of animal is not necessary, specially if it is a specimen mounted for exhibition, not scientific collection.

Figures 4-10 – In addition to photographs of dissected muscles please provide interpretative drawing: the outlines for the muscles as part b of given figure.

Figures lack of scale bare. Please provide scale bar to all your figures. It is mandatory for anatomical drawings.

Table 2. – Why your “table 2” is a graph? Please provide real table with mean +- SD (or better Standard Error, or both) and sample size (N).

Discussion

You have to address the problem of intraspecific variation. Unfortunately, you studied only single specimens, thus you have no data on the within-species variation. However, in this species there is significant sexual size dimorphism in body size, and males wrestle when mate-contest using their arms to hold the rival.

This section is a good place to compare your results to other species/studies. However, first do serious literature search…

I find your discussion speculative. You do not refer to other studies in this section.

Lines 328-329: Again, what is the link between limb musculature and species conservation? I got impression, that authors find conservation issues important and try to put their study in this context, no matter what.

Line 340: “scaly bone”? What do you mean?

Lines 340-347 – I do not understand the link between the centre of gravity and communication in those lizards. More, “The specific anatomy of the skeleton with only one middle ear ossicle may indicate the involvement of vibrational communication or the ability to detect ground seismic changes” – any support for this statement? Most of tetrapods (except of mammals) have single middle ear ossicle. Using forelimb for sound (vibration) transmission is known in some amphibians. But here? Is there any evidence that Komodo dragons use such sound detection? If yes – please provide reference.

Lines 348-362: This is the most interesting and relevant part of your discussion. However, you should address this to other species and point differences and similarities. The last sentence of this paragraph is not necessary.

Line 361: (…) fibres well as (…) – shouldn’t be (…) fibres as well as (…)

 Conclusions

Lines 367-368: There is nothing about evolutionary changes nor intermediate forms in your paper, thus such sentence is irrelevant.

Lines 368-374: Please delete. This paper in about anatomy not conservation and there is no place for sad stories about negative impact of habitat lost on this species.

References

Sorry, but it is bit messy…

You refer to several sources that are popular not scientific, peer-reviewed papers, e.g. web page of American Museum of Natural History (their ref. [4]) Please avoid such sources. Instead do serious literature search. I suggest to look for papers (and books) of such authors like Eric Pianka or Walter Auffenberg.

In the references list the source no. [1] is missing. I think all references are shifted “+1”, “+2” or sometimes “+3” and I find several inconsistences in the referring to literature. For example, lines 89-92 refers to paper of Hocknull et al. listed in the references under [10], but in the main text authors refer to [9], which in turn is Ciofi & DeBoer paper, and so on. This must be fixed. Please also check carefully spelling, i.e. Hocknul (line 89) or Hocknull? Moreover, the same paper by Hocknull et al. is listed twice, under [10] and [16].

Line 99 – you refer to double listed Hocknull paper [16], but I think you meant De Buffrenil et al. paper listed under [18], right?

Lines 162-166: you refer to [26]:

There is lot of spelling mistakes in this section. Please correct “Journal of anatomy” to Journal of Anatomy (lines 458, 460, 463, 468,

[27] In this case Biological Reviews (not reviews) is enough.  

Line 456: change “Anatomical rekord” to “Anatomical Record”

Line 458: use italics for Alligator mississippiensis

 Supplementary Materials: I don’t understand this. It seems You doubled all tables and figures that are embedded in the main text into supplementary material. Supplementary material is exclusively for the data, figures, tables etc. that are not presented along with the main text of the paper.

Here id a non-exhaustive list of articles that might be useful for you to revise your paper. Most of them is mandatory.

Akita K. (1992). An anatomical investigation of the muscles of the pelvic outlet in Iguanas (Iguanidae Iguana iguana) and Varanus (Varanidae Varanus (dumerillii)) with special reference to their nerve supply. Annals of Anatomy - Anatomischer Anzeiger 174: 119-129. https://doi.org/10.1016/S0940-9602(11)80328-6

Bishop PJ, Wright MA, Pierce SE. (2021). Whole-limb scaling of muscle mass and force-generating capacity in amniotes. PeerJ 9:e12574 https://doi.org/10.7717/peerj.12574

Christian A., Garland T. (1996). Scaling of Limb Proportions in Monitor Lizards (Squamata: Varanidae). Journal of Herpetology, 30(2), 219–230. https://doi.org/10.2307/1565513

Cieri R. L. (2018). The axial anatomy of monitor lizards (Varanidae). Journal of Anatomy 233: 636-643.

Collar D. C., Schulte J. A., Losos J. B. (2011). Evolution of extreme body size disparity in monitor lizards (Varanus). Evolution 69: 2664–2680. http://www.jstor.org/stable/41240851

Conrad J. L. (2015). Skeletons of the Little-Known Palawan Monitor, Varanus palawanensis (Squamata: Varanidae). Journal of Herpetology 49: 485-490

de Jong J.K. (1927). Varanus komodoensis, Ouwens, Annals and Magazine of Natural History, 19:113, 589-591, DOI: 10.1080/00222932708655542

Fahn-Lai P, Biewener AA, Pierce SE. (2020). Broad similarities in shoulder muscle architecture and organization across two amniotes: implications for reconstructing non-mammalian synapsids. PeerJ 8:e8556 https://doi.org/10.7717/peerj.8556

Gadow H. (1881). Beiträge zur Myologie der hinteren Extremität der Reptilien. Morp Jb

Granatosky M.C. (2020). Testing the propulsive role of m. peroneus longus during quadrupedal walking in Varanus exanthematicus. J. Exp. Zool. Part A. 333: 325-332. DOI10.1002/jez.2361

Haines RW. (1950). The flexor muscles of the forearm and hand in lizards and mammals. J Anat.84(Pt 1):13-29.

Haines RW. (1952). The shoulder joint of lizards and the primitive reptilian shoulder mechanism. J Anat. 86(4):412-422.

Jenkins F. A. Jr., Goslow G. E. Jr. (1983). The functional anatomy of the shoulder of the savannah monitor lizard (Varanus exanthematicus). J. Morph. 175: 195-216. https://doi.org/10.1002/jmor.1051750207

LandsmeerJ. M. F. 1983: The mechanism of forearm rotation in Varanus exanthematicus. J. Morphol. 175: 119-130. DOI10.1002/jmor.1051750202

Padian, K., Olsen, P. E. (1984). Footprints of the Komodo Monitor and the Trackways of Fossil Reptiles. Copeia, 1984(3), 662–671. https://doi.org/10.2307/1445147

Pianka E. R. (1995). Evolution of Body Size: Varanid Lizards as a Model System. The American Naturalist, 146(3), 398–414. http://www.jstor.org/stable/2463114

Pianka, E. R. (1995). Evolution of Body Size: Varanid Lizards as a Model System. The American Naturalist 146: 398–414. http://www.jstor.org/stable/2463114

Pontzer H. (2007) Effective limb length and the scaling of locomotor cost in terrestrial animals. J. Exp. Biol. 210: 1752-1761. https://doi.org/10.1242/jeb.002246

Rieppel O., Grande L. (2007). The anatomy of the fossil varanid lizard Saniwa ensidensis Leidy, 1870, based on a newly discovered complete skeleton. Journal of Paleontology 81: 643–665. doi: doi.org/10.1666/pleo0022-3360(2007)081[0643:TAOTFV]2.0.CO;2

Ritter D. (1995). Epaxial muscle function during locomotion in a lizard (Varanus salvator) and the proposal of a key innovation in the vertebrate axial musculoskeletal system. J Exp Biol 198 (12): 2477–2490. https://doi.org/10.1242/jeb.198.12.2477

Russell A. P., Bauer A. M. (2008). The appendicular locomotor apparatus of Sphenodon and normal-limbed Squamates. In: Gans C., Gaunt A. S., Adler K. (eds.) Biology of the Reptilia, vol. 21, Morphology I, The Skull and Appendicular Locomotor Apparatus of Lepidosauria. Society for the Study of Amphibians and Reptiles, pp: 1-466.

Taylor G, Bonney V. (1905).On the Homology and Morphology of the Popliteus Muscle: A Contribution to Comparative Myology. J Anat Physiol. 40(Pt 1):34-50. https://www.ncbi.nlm.nih.gov/pmc/articles/PMC1287337/

Young B. A., Magon D.K., Goslow G. E. Jr. (1990). Length-tension and histochemical properties of select shoulder muscles of the savannah monitor lizard (Varanus exanthematicus): Implications for function and evolution. J. Exp. Zool. 256:63-74. https://doi.org/10.1002/jez.1402560109

Young, B. A. (1988). The Subclavian Loop of Varanus salvator. Copeia, 1988: 1029–1034. https://doi.org/10.2307/1445728

Round 2

Reviewer 1 Report

Please, add to References "if you want", but believe it will improve the manuscript this work Grassé P-P, Traité de zoologie: anatomie, systematique, biologie, Tome XVI Mammiféres, teguments, squelette. Premier fascicule. Masson et Cie.1967. pp. 778. to explain better the "ulnar patella".

Also, please, review Figure 11. Os humeri

Into image C. Below arrow "Os humeri" insert an arrow to the Head  humeri (proximal)
Above "Os humeri" add arrow + "Condylus humeri (distal).
Because if you have inverted the position of the os humeri, readers cannot understand where is the "ulnar patella" or in distal or in proximal position.

Reviewer 2 Report

 Dear Authors,

I read carefully your paper and your response to my previous comments. I find your manuscript imprved, however I still think it needs some reworking. 

My general impression is this paper aims to be both for veterinaries and comparative biologist (like me) and both areas require little bit different approach. Thus there are different points of view expressed by me and by the second reviewer (probably more vet-oriented). The paper suffers from such approach, since both evolutionary and comparative anatomists as well as veterinaries are not satisfied with organization and presentation of the data. You have to decide what is your paper expected audience.

There is still lot of out of topic digressions through Introduction. I suggest focusing on only forelimb anatomy of varanids and those aspects of its biology that are strictly related to.

Also, you try to discuss lot of evolutionary issues, however, there is not enough evidence in your data (and not enough taxonomic coverage) to justify such conclusions. From evolutionary biologists perspective, there is no need to compare squamate to mammals or birds, as all belongs to different evolutionary lineages. If you found any convergences between any mammal species and monitor lizards, then it could acceptable, but at this moment it is not. The same is true for Varanus – bird comparison. Forearm of bird (wing) works different than lizard arm, and there is no similarities in locomotion, nor they are closely related; they diverged c.a. 280 m.y.a. and birds undergo very deep reorganization of body plan.

I also strongly recommend very careful language editing. Although English is not my first language, I found several linguistic problems, sometimes direct translation of Polish words to English. Like “…lines of organisms with a specialized limb…” should be „lineages …” (I suppose intended meaning was “linie organizmów o wyspecjalizowanych koÅ„czynach”). There is also lot of misspelings and typos, specially in the references. Also – the references are not properly formatted. In some cases you use full author names, in other only initials; In the articles titles in some cases all words starts with capital letters in other no. Please check carefully the journal titles.

“Infraorder Anguimorpha [4] of the family Varanidae (monitor lizards) [5].” – This sentence does not make sense. Moreover, Varanidae belongs to Anguimorpha: infraorder is wider category than family; Anguimorpha covers 8 families: Anguidae, Anniellidae, Diploglossidae, Xenosauridae, Helodermatidae, Shinisauridae, Lanthanotidae and Varanidae (no need to list them in the text, I put them here as example).

“Among them, there are lines (…)” – please change lines to lineages.

(ka) - ? what do you mean?

It is a species that represents gigantism in the animal world, confirmed by morphological and genetic studies [18]. – Yes, it is example of gigantism, but how it is confirmed by genetic studies?

“Hocknull S. et al. took measurements of the fossilised and modern Komodo humerus (…)” – shouldn’t be “(…)Komodo dragon humerus (…)”?

“The arrangement of the ulna and two pea— shaped wrist bones, compared to the Water dragon, give it greater flexibility of movement” – Please rearrange to ” The arrangement of the ulna and two pea— shaped wrist bones, give it greater flexibility of movement compared to the Malayan water dragon Varanus salvator Laurenti, 1768”; I suppose the “water dragon” (water monitor?) you refer to is Varanus salvator, however, agamid lizards Physignathus are also known as water dragons. To avoid confusion please include formal taxon name when first mention.

"Many controversies have arisen around the phylogeny of Squamates [21,26], although more and more research allows to distinguish between the crown squamata in their main groups." - The references are deQueiroz et al. papers from 1988 (more than 30 years ago!) and since this time significant progess has been made in resolving animal phylogeny! Please check recent papers (e.g. Pyron et al. 2013: BMC Evol Biol. 13:93. doi: 10.1186/1471-2148-13-93; Simões and Pyron 2021: Bulletin of the Museum of Comparative Zoology, 163(2), 47-95 and other recently publshed papers). Second, this sentence mess concept of crown-group. What do you mean by "crown squamata in their main group"?

“Some species of Varanidae have developed bipedalism which has allowed them greater maneuverability [29], and these changes have been dated to 110 million years ago [30] And it may have had an effect on acceleration. Facultative bipedalism is limited in Varanidae to Varanus gouldii, panoptes and giganteus. [31]” – Please rearrange to: “Some species of varanids, Varanus gouldii, V. panoptes and V. giganteus, have developed facultative bipedalism which has allowed them greater maneuverability and may have had an effect on acceleration [29, 31], and these changes have been dated to 110 million years ago [30].” – please add “facultative” since there is no single species of monitor lizards that is fully bipedal; it is always quadrupedal way of locomotion and in some occasions they run for short distance on two legs. When you list several species of the same genus, you should use full species name or at least the first letter of the generic name plus full species epithet.

“Malayan water monitor (Varanus salvator Laurenti, 1768)” – please compare to one of my previous comments. If the “water dragon” you mentioned above was V. salvator, then you can use only common name here. But please make sure if it should be water dragon or water monitor.

"may be an example in which the biochemistry of muscle work is better explained; they store metabolically produced carbon dioxide as plasma dicarbonate, removing lactic acid from the blood very efficiently. The transition from aerobic to anaerobic metabolism, anatomy and muscle [39]—blood biochemistry is crucial not only due to anatomical differences but also in anaesthesia, manual administration of the drug to the limb or tail is suggested and possible metabolic effects must be taken into account. In V. komodoensis it is recommended to administer drugs intramuscularly to the forelimb for anesthesia induction or recovery (midazolam, ketamine or tiletamine-zolazepam) [40, 41]." - I don't think it fits to the scope of your paper. I suggest to delet this part. 

“Such variety of limbed and limbless forms is observed in all Anguimorpha infraorder” – please rephrase to: “Such variety of limbed and limbless forms is observed across Anguimorpha”.

“Histological preparations were made” – What kind of staining did you use? What was  histological sections thickness? Did you use paraffin of cryostat sections. Please specify the histological protocol.

“Unfortunately, the literature on the anatomy of the Komodo dragon’s detailed anatomy is rather insufficient. In general, the research collected in the literature on this species can be divided into two stages; from 1912 to 1970, and after. The first one focused on expeditions, taxonomy of the species, measurements and obtaining data from the local population and individuals for breeding in zoos, and the consequent difficulties in breeding. In the second, attempts were made to determine the occurrence of the species in individual areas of the Indonesian islands and research expeditions were organised to evaluate the behaviour. After 1971, interest in monitor lizards increased, knowledge of growth relationships, blood test data, sex determination, social behaviour and chromosomal evolution of genes was gained [2,14,40,51,55,]. Nevertheless, the literature on the subject is still very limited, often local in scope. Species expert Surahya S., a biologist at Gadjah Mada University, mentioned that he has not found a successor to his research (nasional.kompas.com [59]). Advanced muscle measurements were performed by the Royal Veterinary College (UK) on a male deceased at London Zoo by Hutchinson J., Stoll A., Allen V., Regnault S. and Macaulay S., but data on the muscles of the thoracic limb haven’t been published (theatlantic.com [60])” – this part belongs to the Introduction section.

“Anatomical data on crocodiles (Crocodylidae) and their relatives (Alligatoridae) [48—51,66], Iguana iguana [67,68] as well as comparative data on other tetrapods [25,49,69]” – Here is lacking something. Anatomical data was what? Sentence must have the verb.

“The total length of the forelimb was 28.7 cm. The length of the hand (from the last phalanx, base of the claw to the head of the ulna) was 6.9 cm, forearm 9.8 cm and shoulder 12 cm. The animal's SLV was 93.6 cm and the length of the head was 18 cm” – only this part of the entire paragraph belongs to the Results section. The other part is not necessary.

“no significant analogy to the limb of birds was found [62,63]” – Come on! Birds are highly specialized lineage of Archosauria. Their locomotion (flight!) require deep reorganization of musculature. There is no point to justify comparison of varanids to birds! Moreover, comparative studies of birds and other tetrapods has been done several times, years ago (can be dated back to Thomas Henry Huxley!) and there is no need to repeat!

„Significantly developed coracoids (…)” – Do you mean “strongly developed coracoids…”?

“In V. komodoensis there was no reduction of the phalanges from 2:3:4:5:3 to 2:3:3:3:3 as in mammal-like reptiles and mammals [76]” – This is nothing new. Please delete this sentence. Here you try to compare a lizard with mammals or mammal-like reptiles, however, these are distantly related and you not expect nor discuss here any possible convergences among these groups. Such remarks would be justified if large number of taxa representing various lineages were analyzed within phylogenetic framework.

“The anatomy of these animals shows many similarities but also special features, for example those that may directly result from pedomorphism [27].” – Which animals do you mean in the context of pedomorphosis? Snakes? Amphisbaenians (Diplometopon or Diplomethoyjon – see below). Also, should be “… from pedomorphosis” not pedomorphism.

“Figure 13. Selected morphometric measurements of muscle fibres with an indication of the average width (µm) and (+/- SD) standard deviation (Microsoft Corporation (2018). Microsoft Excel). Each sample was based on 100 morphometric measurements” – In my opinion table would be better to present mean and SD values. Here it is hard to extract what was actually the mean and what is value of standard deviation.

Results and Discussion:

The Results section is much better organized comparing previous version of your paper. However, in my opinion in many parts it is mixed with Discussion rather, than Results parts. For example:

“The presence of sesamoid (,,ulnar patella”) (11), was confirmed by the performed dissection. Located between the condylus ulnaris of the humerus and the olecranon of the ulna, in the ligament of this joint. It may indicate an increased biomechanical advantage (an enhanced anti-gravity lever system of the elbow joint); it was already described in the forelimb of many taxa, especially lizards [75].”

The first two sentences describes the sesamoid, and this is clearly “Results”. However the latter part of this paragraph [“It may indicate an increased biomechanical advantage (an enhanced anti-gravity lever system of the elbow joint); it was already described in the forelimb of many taxa, especially lizards [75]]” should be moved to discussion. And this mix is very common in your Results section.

Typo and formatting issues:

Do not use italics for latin names other than generic, species and subspecies epithets. Thus, Alligatoridae or Crocodylidae are family names (end with -idea). Here is a list of italicized (or not-italicized) words to be corrected:

p. 2: “representative of the Indonesian archipelago and belongs to the Toxicofera clade of…” Toxicofera

P. 3: “We know, for example, that in Varanus…” Varanus

“General commonalities were found in the neck myology of Varanid lizards and snakes (Trimeresurus) and Diplomethoyjon…” Trimeresurus. Btw. What is Diplomethoyjon? I have never meet such taxon in the literature. I also tried to find this name in some databases, but failed (and dr Google failed too). Do you mean Diplometopon? This is an amphisbaenid species (Diplometopon zarudnyi).

p. 4: “In the entire Varanid family…” Varanid

“What has already been conducted, e.g. in crocodiles (Crocodylidae) and relatives (Alligatoridae).” Crocodylidae, Alligatoridae

p. 22. “Ciofi, C.; Puswati, J.; Winana, D.; de Boer, M.E.; Chelazzi, G.; Sastrawan, P. Preliminary Analysis of Home Range Structure in the Komodo monitor, Varanus komodensis.” Varanus komodoensis

p. 23. “The Anatomical R ecord” – please remove the space in Record (It appears several times).

“Monitoring muscle over Tyree orders of magnitude: Widespread positive allometry in the pectoral girdle of varanid lizards (Varanidae).” Should be “…three orders…”

p. 24. “The Anatomical Rekord” – should be “Record”.

67. Menezes Freitas, L.; Sabec-Pereira, D.K.; Fernando Pereira, K.; Paulo Dos Santos, O.; Lim, F.C. Muscular anatomy of the pectoral girldle and forelimb of Iguana i. iguana (Squamata: Iguanidae). Bioscience Journal 2017, 33(5), pp. 1284—1294.

68. Freitas, L.M.; Pereira, D.K.S.; Pereira, K.F.; dos Santos, O.P.; Lima, F.C. Muscular Anatomy of the Pectoral Girdle and Forelimb of Iguana i. iguana (Squamata: Iguanidae). Journal of Biosciences 2017, 33(5), pp. 1284—1294.

These two references have the same titles, year, volume and pagination data, but differs in authors list and journal names (Bioscience Journal vs. Journal of Biosciences). I suppose only one is correct…

“Ecological allometrie and niche use Dynamics cross Komodo dragon onkogeny” – change to “Ecological allometries and niche use dynamics across Komodo dragon ontogeny”
